# CTCF is a barrier for 2C-like reprogramming

Teresa Olbrich[1], Maria Vega-Sendino[1], Desiree Tillo [2], Wei Wu [1], Nicholas Zolnerowich[1], Raphael Pavani[1], Andy D. Tran [3], Catherine N. Domingo[1], Mariajose Franco[1], Marta Markiewicz-Potoczny [1], Gianluca Pegoraro[4], Peter C. FitzGerald [5], Michael J. Kruhlak[3], Eros Lazzerini-Denchi [1], Elphege P. Nora [6,7], André Nussenzweig[1] & Sergio Ruiz [1]✉

Totipotent cells have the ability to generate embryonic and extra-embryonic tissues. Interestingly, a rare population of cells with totipotent-like potential, known as 2 cell (2C)-like cells, has been identified within ESC cultures. They arise from ESC and display similar features to those found in the 2C embryo. However, the molecular determinants of 2C-like conversion have not been completely elucidated. Here, we show that the CCCTC-binding factor (CTCF) is a barrier for 2C-like reprogramming. Indeed, forced conversion to a 2C-like state by the transcription factor DUX is associated with DNA damage at a subset of CTCF binding sites. Depletion of CTCF in ESC efficiently promotes spontaneous and asynchronous conversion to a 2C-like state and is reversible upon restoration of CTCF levels. This phenotypic reprogramming is specific to pluripotent cells as neural progenitor cells do not show 2C-like conversion upon CTCF-depletion. Furthermore, we show that transcriptional activation of the ZSCAN4 cluster is necessary for successful 2C-like reprogramming. In summary, we reveal an unexpected relationship between CTCF and 2C-like reprogramming.

[1] Laboratory of Genome Integrity, CCR, NCI, NIH, Bethesda, MD, USA. [2] Genetics Branch, CCR, NCI, NIH, Bethesda, MD, USA. [3] Laboratory of Cancer Biology and Genetics, CCR, NCI, NIH, Bethesda, MD, USA. [4] Laboratory of Receptor Biology and Gene Expression, CCR, NCI, NIH, Bethesda, MD, USA. [5] Genome Analysis Unit, CCR, NCI, NIH, Bethesda, MD, USA. [6] Cardiovascular Research Institute, University of California San Francisco, San Francisco, CA, USA. [7] Department of Biochemistry and Biophysics, University of California San Francisco, San Francisco, CA, USA. ✉email: sergio.ruizmacias@nih.gov

Totipotency is defined as the ability of a single cell to generate all cell types and is found in zygotes and 2-cell (2C) embryos[1,2]. As development proceeds, embryonic cells progressively restrict their developmental potential. Embryonic stem cells (ESC) isolated from the inner cell mass (ICM) of blastocysts are defined as pluripotent since they lack the ability to differentiate into extra-embryonic tissues[1,2]. Interestingly, a rare (~1–2%) transient population of cells with totipotent-like potential was identified within ESC cultures[2–4]. This cell population expresses high levels of transcripts detected in 2C embryos, including a specific gene set regulated by endogenous retroviral promoters of the MERVL subfamily[2–4]. At the 2C embryonic stage, these retroviral genetic elements are re-activated and highly expressed when the zygotic genome is first transcribed and quickly silenced after further development. Based on this specific feature, retroviral promoter sequences (LTR) have been used as a reporter system to genetically label 2C-like cells in vitro to study their behavior and properties[2–4]. Previous studies have shown the role of different genes and pathways in converting ESC to a 2C-like state in vitro[3,4]. Indeed, expression of the transcription factor DUX in ESC is necessary and sufficient to induce a 2C-like conversion characterized by similar transcriptional and chromatin accessibility profiles, including MERVL activation, as observed in 2C-blastomeres[5–7]. This reprogramming cell model has been instrumental to study the molecular mechanisms that regulate the acquisition and maintenance of totipotent-like features. DUX belongs to the double homeobox family of transcription factors exclusive to placental mammals[8] and is expressed exclusively in the 2C embryo[5–7]. Interestingly, DUX knockout mice revealed that DUX is important but not essential for development, suggesting that additional mechanisms regulate zygotic genome activation (ZGA) and the associated totipotent state in vivo[9,10].

In this study, we demonstrate that the zinc-finger binding protein CTCF, involved in regulating the higher-order organization of chromatin structure, is a barrier in pluripotent cells for 2C-like reprogramming.

## Results

**2C-like conversion correlates with DNA damage and cell death.** To explore new molecular determinants regulating totipotency, we generated ESC carrying a doxycycline (DOX)-inducible DUX cDNA (hereafter, ESC$^{Dux}$)[11]. Upon DOX activation we detected the expected expression of *Dux* and its downstream ZGA-associated genes (Supplementary Fig. 1a, b). In addition, ESC$^{Dux}$ containing an LTR-RFP reporter showed reactivation of MERVL sequences after DOX induction (Supplementary Fig. 1c, d). Over-expression of DUX triggers toxicity in myoblasts[12]. However, whether sustained expression of DUX leads to cell death in ESC has not been explored thoroughly. We observed that DUX expression induced cell death in a dose and time-dependent manner and correlated with the extent of 2C-like conversion (Fig. 1a). Indeed, live cell imaging of DOX-treated ESC$^{Dux}$ expressing H2B-eGFP showed efficient cell death in cells asynchronously converting to a 2C-like state (Supplementary Fig. 2a, Supplementary Movie 1). Interestingly, accumulation of DOX-induced ESC$^{Dux}$ in the G1 and G2 phases of the cell cycle along with a decrease in DNA replication preceded cell death (Fig. 1b, Supplementary Fig. 2b). To exclude that these effects were due to supra-physiological levels of DUX, we analyzed the unperturbed subpopulation of ESC that spontaneously undergoes a 2C-like conversion[3]. These endogenous 2C-like ESC were also characterized by G2 accumulation, decreased DNA replication, and overt spontaneous cell death following 2C-like conversion (Supplementary Fig. 2c–e and Supplementary Movie 2). In support of these observations, the activation of the transcriptional 2C program during ZGA following the first cleavage in fertilized zygotes is accompanied by an extremely long G2 phase (around 12–16 h)[13,14].

We next examined whether decreased DNA replication and G2 accumulation in DOX-treated ESC$^{Dux}$ correlated with elevated levels of replication stress (RS). Indeed, we observed that sustained expression of DUX leads to DNA-damage, revealed by the increased levels of the RS markers KRAB-associated protein 1 (KAP1) and phosphorylated H2AX (γH2AX) in a dose and time-dependent manner (Fig. 1c, d). We also detected higher levels of γH2AX in endogenous 2C-like ESC (Supplementary Fig. 2f). Thus, the decrease in DNA replication and elevated levels of γH2AX observed in 2C-like ESC suggested that RS could underlie the increased levels of DNA damage and reduced cell viability in these cells. Accordingly, increasing RS levels by using an ATR inhibitor showed an additive effect of RS and DUX expression on DNA damage (Supplementary Fig. 3a). We hypothesized that DUX-mediated increased transcription of the 2C-associated genes could be, at least partially, responsible for the RS observed. To test our hypothesis, we performed ChIP-seq analyses to detect chromatin-enrichment of the single-strand DNA (ssDNA) binding protein RPA. RPA accumulates on ssDNA upon transcription-replication conflicts resulting in replication fork stalling and DNA damage during RS[15,16]. We examined RPA accumulation in the one hundred most upregulated genes upon DUX expression and observed increased RPA enrichment near the DUX binding site and the transcription start site in DOX-treated ESC$^{Dux}$ compared to untreated ESC$^{Dux}$ (Fig. 1e, f). Similarly, we also detected RPA accumulation in re-activated MERVL sequences (Supplementary Fig. 3b, c). These results suggested that transcription-replication conflicts might arise at DUX-induced highly transcribed genes and repeats reinforcing the idea that induction of a 2C-like state by DUX in ESC is associated to RS-mediated DNA-damage. Interestingly, it has been recently shown that cultures of ESC treated with RS agents such as hydroxyurea, UV-light or cisplatin showed elevated expression of specific genes found in 2C embryos and 2C-like cells[17,18].

Combined with our results, these observations suggest the existence of an intertwined relationship between DNA damage and the 2C-like transcriptional program in ESC.

**DUX-induced DNA damage localizes at CTCF binding sites.** We next sought to investigate whether DUX-induced RS and DNA damage could result in DNA breakage. To explore this possibility, we performed END-seq, a highly sensitive method to detect recurrent DNA ends genome-wide at base-pair resolution[19]. DUX-expressing ESC showed de novo accumulation of END-seq signal at specific genomic locations compared to untreated ESC$^{Dux}$. Indeed, a total of 1539 END-seq peaks overlapped between two independent ESC$^{Dux}$ clones (Supplementary Data 1–3). Moreover, the type of lesion (double or single strand DNA break) at each site, showed high correlation when both ESC$^{Dux}$ clones were compared (Supplementary Fig. 4a–c). More than 25% of the END-seq peaks localized within a 10 kb distance from a DUX binding site (hypergeometric test p-value: 1.62E-32, computed using the average END-seq peak length and the size of the mouse genome). Furthermore, 16% of the 1220 genes associated by proximity to END-seq peaks, including well-known 2C genes, were strongly upregulated by DUX (hypergeometric test p-value: 1.75E-30 using a gene set population size of 27,590 genes[5]; Supplementary Fig. 4d and Supplementary Data 4 and 5). These results showed that DUX-induced 2C-like conversion reproducibly generated DNA lesions in specific genomic regions associated with DUX-induced transcription. We next asked

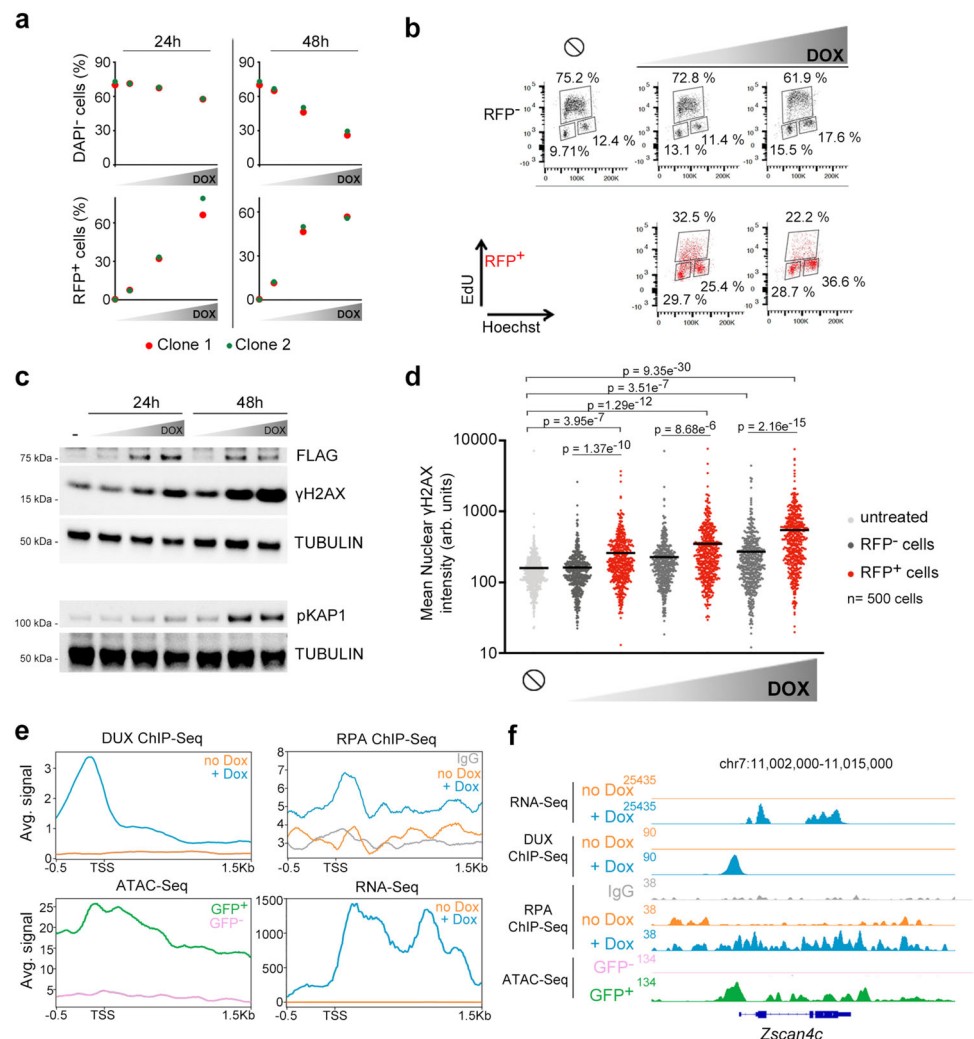

**Fig. 1 Induction of 2C-like features in ESC correlated with DNA damage and cell death. a** Plots showing the percentage of alive (DAPI⁻) and RFP⁺ cells from LTR-RFP reporter ESC^DUX treated with increasing doses of DOX (50, 150 and 300 ng/mL) for the indicated time points. Data was collected by flow cytometry. **b** Flow cytometry analysis of the cell cycle distribution in untreated or 150 and 300 ng/mL DOX-treated LTR-RFP reporter ESC^DUX for 24 h. Percentages for each phase of the cell cycle are included. **c** Western blot analysis of the indicated proteins performed in ESC^DUX treated with different doses of DOX (150, 300 and 600 ng/mL) for the indicated time points. Expression of DUX was monitored by its FLAG-tag. Tubulin levels are shown as a loading control. **d** High-throughput imaging (HTI) quantification of γH2AX in LTR-RFP reporter ESC^DUX treated with different concentrations of DOX (150, 300 and 600 ng/mL) for 24 h. Center lines indicate mean values. ∅ = No treatment. n = 500; p-values are shown from two-tailed unpaired t-tests. In **b** and **c**, cells were split in RFP⁻ or RFP⁺. In **a–d**, two independent experiments with similar results were performed with two different clones, but one representative experiment is shown. **e** Average read density plots (RPKM) showing RPA, DUX occupancy and RNAseq read counts from untreated or DOX-treated DUX expressing ESC (our data and ref. [5]). ATACseq signal is shown from GFP⁻ or GFP⁺ cells sorted DOX-treated LTR-GFP reporter ESC^DUX[5]. Plots were generated using the 100 most upregulated genes upon DUX expression in ESC[5]. TSS = Transcription start site. **f** Genome browser tracks showing RPA, DUX occupancy, RNAseq read counts and ATACseq signal in the region surrounding Zscan4c.

whether these regions shared any feature that could explain the reiterative DNA damage on them. Thus, we performed a transcription factor motif enrichment analysis using our END-seq peak dataset and found the CTCF binding motif as one of the most significant (Fig. 2a). Using published CTCF ChIP-seq datasets in ESC[20], we confirmed that around 50% of the END-seq peaks were occupied by CTCF (Fig. 2b, c, Supplementary Fig. 4c, e, f and Supplementary Data 6). Moreover, these sites were also enriched in SMC1 and SMC3[21], components of the cohesin ring-like protein complex (Fig. 2b, c, Supplementary Fig. 4e).

The transcription factor CTCF is a zinc-finger binding protein involved in chromosome folding and insulation of topologically associated domains (TADs)[22]. Based on the observed CTCF-associated DNA damage in DOX-induced ESC^Dux, we speculated that CTCF might represent a barrier for the reprogramming to a

2C-like state. This idea was supported by two observations. First, cohesin depletion in differentiated cells facilitates reprogramming during somatic cell nuclear transfer by activating ZGA[23]. Second, totipotent zygotes and 2C embryos are characterized by chromatin in a relaxed state associated with weak TADs[24,25]. Following fertilization, development is accompanied by a progressive maturation of higher-order chromatin architecture[24,25]. Interestingly, increasing levels of CTCF during human embryonic development are required for the progressive establishment of TADs[26]. Similarly, we also observed a steady increase in the levels of CTCF during development in mouse embryos (Supplementary Fig. 5). Thus, we first sought to examine the CTCF binding landscape in 2C-like cells by native Cut&Run sequencing. For this, we used LTR-RFP reporter ESC^DUX to first induce 2C-like conversion, and then, sort RFP⁺ and RFP⁻ cells

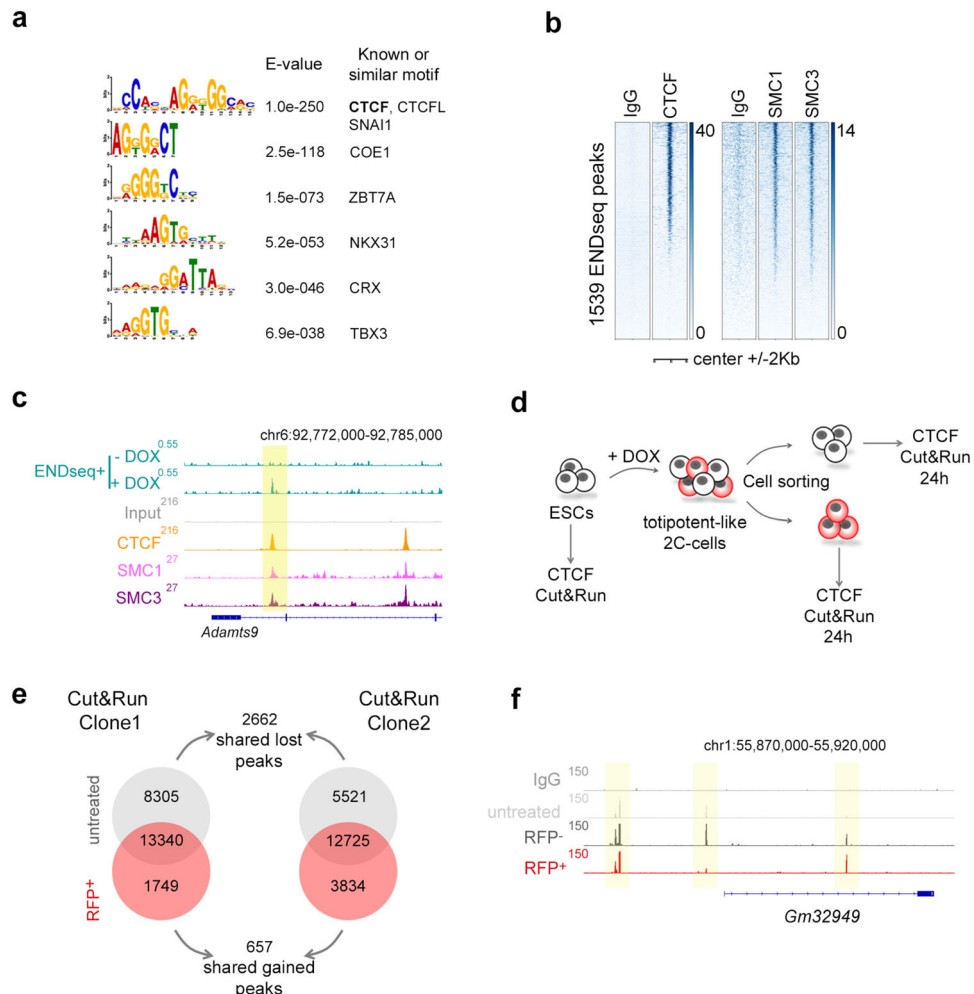

**Fig. 2 DUX expression is associated with DNA damage at a subset of CTCF binding sites. a** Results from a motif enrichment analysis performed on the set of overlapping 1539 END-seq sites identified in DUX-expressing ESC. **b** Heatmaps showing CTCF[20], SMC1 and SMC3[21] occupancy at the set of 1539 END-seq sites. **c** Genome browser tracks showing END-seq signal in untreated and DOX-treated ESC[DUX] at the indicated genome location. In addition, CTCF[20], SMC1 and SMC3[21] occupancy in ESC is shown. END-seq peak is highlighted. **d** Schematic representation of the experiment performed. **e** Venn diagram showing the number of CTCF peaks detected in untreated and RFP+ sorted cells from two independent DOX-treated ESC[DUX] clones. The number of overlapping shared gained and lost CTCF peaks is also shown. **f** Genome browser tracks showing CTCF occupancy at the indicated genome location in one representative ESC[DUX] clone. Input (IgG) is shown as a background reference control. Note how in the region shown, one CTCF peak is gained and one lost in RFP+ cells. In all cases, 600 ng/mL DOX was used.

24 h after DOX induction (Fig. 2d–f). Interestingly, RFP+ cells are characterized by a decrease in the number of CTCF peaks identified (Fig. 2e, f and Supplementary Data 7). Indeed, a total of 2662 and 657 overlapping CTCF peaks in two independent ESC[Dux] clones were lost and gained, respectively, in RFP+ cells compared to untreated ESC[Dux] (Fig. 2e, f). Of note, although RFP− cells did not reprogram to a 2C-like state, they also showed some level of reorganization in their CTCF binding landscape (Supplementary Fig. 6a). These changes were not due to variations in the total levels of CTCF (Supplementary Fig. 4g). In addition, spontaneously converting 2C-like ESC were also characterized by a similar CTCF binding reorganization (Supplementary Fig. 6b–d and Supplementary Data 8).

**CTCF depletion leads to spontaneous 2C-like conversion.** We next asked whether CTCF loss influences the acquisition of 2C-like features. Thus, we used an auxin-inducible degron system to deplete CTCF in ESC[27]. This cell line (ESC[CTCF-AID] hereafter) harbors both *Ctcf* alleles tagged with an auxin-inducible degron

(AID)[28] sequence fused to eGFP. Although CTCF-AID protein levels in ESC[CTCF-AID] are lower compared to untagged CTCF in wild-type cells, ESC[CTCF-AID] showed negligible transcriptional changes as tagged CTCF retains most functionality[27]. To test whether CTCF deletion induces conversion to 2C-like cells we first examined in CTCF-depleted cells the expression levels of the zinc finger protein ZSCAN4, a gene cluster that encodes six ZSCAN4 paralogs (ZSCAN4A, ZSCAN4B, ZSCAN4C, ZSCAN4D, ZSCAN4E, and ZSCAN4F) and three pseudogenes (ZSCAN4-PS1, ZSCAN4-PS2 and ZSCAN4-PS3). ZSCAN4 (considered as a cluster unless otherwise noted) is selectively expressed in 2C embryos and 2C-like ESC[3,29]. Although all ZSCAN4 transcripts are expressed in ESC, ZSCAN4C and ZSCAN4F are the most abundant[18]. Strikingly, ZSCAN4 levels were elevated two days following CTCF depletion and further increased two days later (Fig. 3a and Supplementary Fig. 7a). Indeed, more than 20% of the cells expressed ZSCAN4 three days following CTCF depletion (Supplementary Fig. 7b). Importantly, similar percentages of RFP+ cells were observed in LTR-RFP reporter ESC[CTCF-AID] (Fig. 3b). This percentage decreased upon

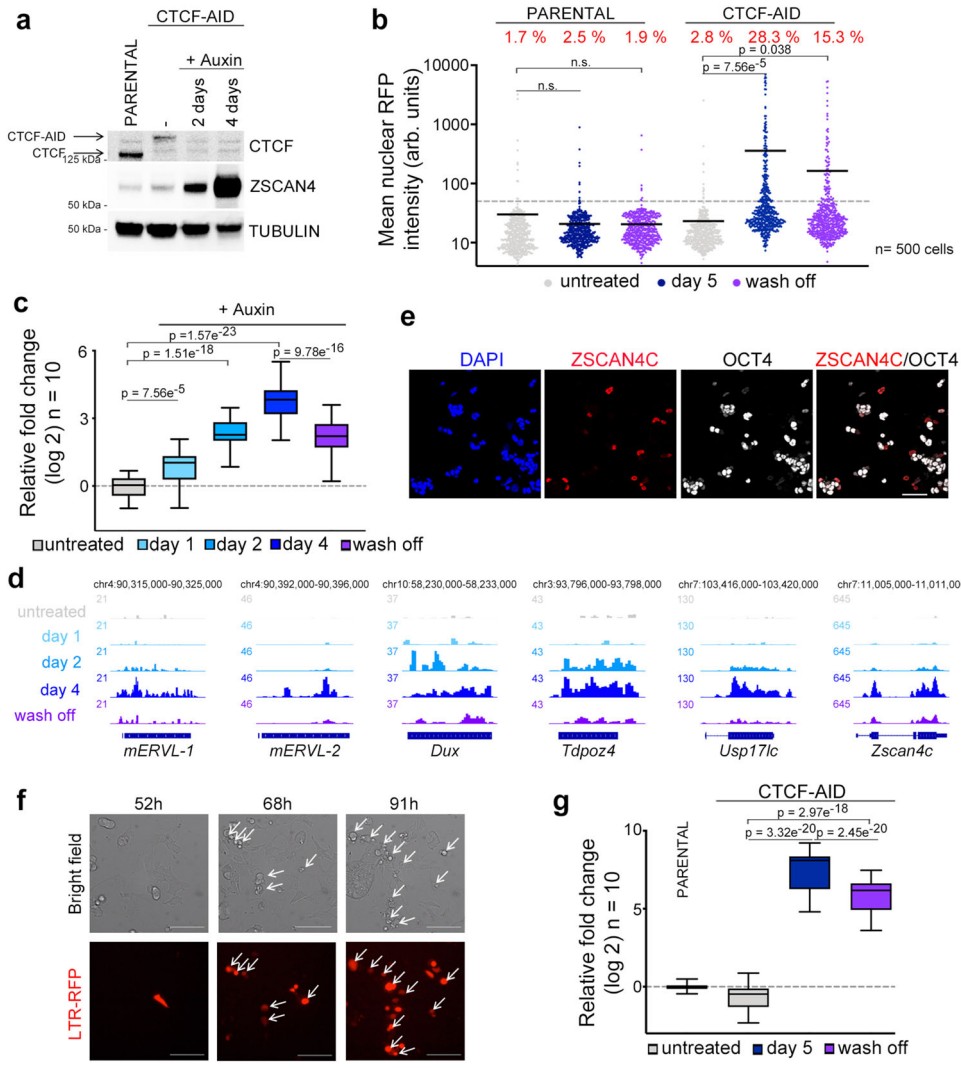

**Fig. 3 Spontaneous 2C-like conversion in CTCF-depleted ESC. a** Western blot analysis performed in parental ESC and ESC^CTCF-AID treated with auxin for the indicated times. Tubulin levels are shown as a loading control. **b** HTI quantification of RFP+ cells in untreated or auxin-treated for five days LTR-RFP reporter parental ESC and ESC^CTCF-AID. RFP+ cells two days after a wash-off following three days of auxin treatment were also quantified. Center lines indicate mean values. Percentages of RFP+ cells above the threshold are indicated. n = 500; p-values are shown from two-tailed unpaired t-tests. n.s. = non-significant. **c** Box and whisker plot showing the relative fold change (log2) expression of ten 2C-associated genes/repeats (DUX, ZSCAN4, ZFP352, TCSTV3, SP110, TDPOZ1, DUB1, EIF1ad8, PRAMEL7 and MERVLs) in ESC^CTCF-AID treated with auxin for the indicated times. Untreated and wash off ESC^CTCF-AID were also included. Data was obtained from RNAseq datasets[27]. **d** Genome browser tracks showing RNAseq RPKM read count at the indicated genes in the same samples as in **c**. **e** Immunofluorescence analysis of ZSCAN4 and OCT4 in ESC^CTCF-AID treated with auxin for 4 days. DAPI was used to visualize nuclei. Scale bar, 100 μm. **f** Representative bright field images (upper panels) from a time lapse experiment performed in auxin-treated LTR-RFP reporter ESC^CTCF-AID. RFP+ cells are shown as they convert over time (lower panels). Time since the addition of auxin is indicated. White arrows indicate 2C-like converted cells undergoing cell death. Scale bars, 100 μm. **g** Box and whisker plot showing the relative fold change (log2) expression of ten 2C-associated genes (same as in **c**) in LTR-RFP reporter ESC^CTCF-AID untreated or treated with auxin for four days and further incubated with auxin or washed-off for additional 18 h (a total of five days) and sorted based on RFP expression. GAPDH expression was used to normalize. For box plots (**c**, **g**), center line indicates the median, box extends from the 25th to 75th percentiles and whiskers show Min to Max values. p values are shown from two-tailed paired t-tests. In **a**, **b**, **e**–**g**, one representative experiment is shown but three (**a**, **b**) and two (**e**–**g**) independent experiments were performed.

restoration of the CTCF levels by washing off auxin (Fig. 3b). Using RNAseq datasets from CTCF-depleted cells at different timepoints, we observed a progressive increase in the expression of genes enriched or exclusively expressed in 2C embryos or 2C-like ESC (Fig. 3c, d). Among these, endogenous MERVL sequences as well as *Dux* were selectively expressed over time upon CTCF depletion (Fig. 3d). We also observed decreased expression of the pluripotent gene OCT4 in ZSCAN4+ auxin-treated ESC^CTCF-AID as described for 2C-like ESC (Fig. 3e)[5]. Furthermore, CTCF-depleted ESC showed transcriptional similarity with DUX-overexpressing ESC (Supplementary Fig. 7c). We

next asked whether expression of DUX, known to efficiently promote 2C-like conversion[5–7], could cooperate to further promote 2C-like reprogramming in auxin-treated ESC^CTCF-AID. By expressing low levels of DUX to limit ESC death and avoid saturation in 2C-like conversion, we observed increased 2C-like reprogramming (Supplementary Fig. 7d). We also examined the effect of HDAC inhibitors in the 2C-like conversion mediated by CTCF depletion. Indeed, histone acetylation has been shown to regulate the expression of ZSCAN4 and other 2C genes[30]. Moreover, the PSPC1-TET2 complex is able to recruit HDAC1/2 to repress the expression of MERVL sequences[31]. Our data

showed that 2C-like conversion was further boosted cooperatively with the use of HDAC inhibitors in auxin-treated ESC[CTCF-AID] (Supplementary Fig. 7e). Finally, we validated these observations by generating additional ESC[CTCF-AID] clonal lines (Supplementary Fig. 7f). Collectively, these results demonstrated that CTCF depletion leads to spontaneous 2C-like conversion in ESC.

We next examined the dynamics of the 2C-like conversion by live cell imaging in LTR-RFP reporter ESC[CTCF-AID]. Reprogramming to 2C-like ESC is asynchronous as ESC convert over time after CTCF depletion (Fig. 3f). Interestingly, we observed that spontaneously converted 2C-like ESC undergo similar cell death as shown for endogenous 2C-like ESC while non-converted ESC divide and do not show overt cell death (Fig. 3f, Supplementary Movie 3). Indeed, although CTCF depletion does not lead to increased DNA damage when the population is considered as a whole[27], CTCF-depleted 2C-like ESC showed increased γH2AX, similar to endogenous 2C-like ESC (Supplementary Fig. 7g). Our data suggested that cell toxicity induced by CTCF-depletion is due to the selective death of the spontaneously converted 2C-like ESC. Finally, we explored whether restoring CTCF expression facilitates the exit from the 2C-like state. For this, CTCF-depleted LTR-RFP reporter ESC[CTCF-AID] for four days were either further incubated with auxin or washed off for an additional 18 h (5 days total) and sorted based on RFP expression. Gene expression analysis showed that restoration of CTCF levels induced a decrease in the 2C-like transcriptional program in 2C-like cells anticipating the exit from the totipotent-like state (Fig. 3g and Supplementary Fig. 8). Collectively, these results demonstrated that CTCF prevents 2C-like conversion.

**Reprogramming roadblocks to 2C-like conversion**. We observed that 2C-like conversion mediated by CTCF depletion is not fully penetrant. Indeed, around 15-25% of CTCF-depleted ESC can successfully undergo 2C-reprogramming within 4 days of depletion. We then asked whether intrinsic heterogeneity within ESC cultures could influence the efficiency of the 2C-like reprogramming. Thus, we established a total of 23 single-cell derived clonal ESC lines from the parental ESC[CTCF-AID] and examined reprogramming dynamics. We determined that the endogenous percentage of 2C-like cells within the cultures of these clonal ESC lines varies from 0.17% to 2.87% (Supplementary Fig. 9a). Interestingly, we found a significant correlation between the final percentage of 2C-like cells observed upon CTCF depletion and the starting percentage in the same clonal ESC[CTCF-AID] line (Supplementary Fig. 9a). This observation reinforces the idea that intrinsic transcriptional and/or epigenetic variability is a determinant of successful 2C-like conversion.

To further support this idea, we explored whether lineage committed cells showed the same 2C-like conversion phenotype. For this, we differentiated ESC[CTCF-AID] to proliferative neural stem cells (NSC[CTCF-AID]) (Supplementary Fig. 9b, c). Auxin-treated NSC[CTCF-AID] did not show increased 2C-associated marker expression or undergo 2C-like reprogramming upon CTCF depletion (Fig. 4a–d and Supplementary Fig. 9d). Collectively, these results suggested that the epigenetic and transcriptional changes taking place during lineage commitment and differentiation toward NSC[CTCF-AID] impose additional roadblocks that prevent 2C-like reprogramming in CTCF-depleted cells.

**ZSCAN4 expression is required for 2C-like reprogramming**. Endogenous emergence of 2C-like cells in ESC cultures is a stepwise process defined by sequential changes in gene expression[32]. ZSCAN4+MERVL− ESC are detected during this process and represent an intermediate step that precedes the full conversion to a 2C-like state[32,33]. Levels of ZSCAN4 progressively increase during 2C conversion prior to the activation of MERVL sequences and the expression of chimeric transcripts[32,33]. Accordingly, we also detected a progressive accumulation of ZSCAN4 in CTCF-depleted ESC starting as early as 24 h after depletion (Fig. 5a and Supplementary Fig. 10a, b). However, upregulation of DUX or MERVL sequences was observed at later timepoints, suggesting that spontaneous conversion upon CTCF depletion followed a similar molecular roadmap as endogenous 2C-like cells. In agreement, we also detected ZSCAN4+mERVL− ESC in early auxin treated LTR-RFP reporter ESC[CTCF-AID] (Supplementary Fig. 10c, d). Importantly, we did not detect changes in the expression level of known regulators of the 2C-like conversion[3,32,34–36] 24 or 48 h after CTCF depletion suggesting that CTCF could directly control the expression of the ZSCAN4 cluster in ESC (Supplementary Fig. 11a). Thus, we asked whether early transcriptional activation of ZSCAN4 in ESC precursors is essential for full conversion to 2C-like cells. Therefore, we infected LTR-RFP reporter ESC[CTCF-AID] with lentiviruses expressing shRNAs targeting ZSCAN4 paralogs (see methods for details) and examined transcriptional dynamics and 2C-like conversion upon CTCF removal. We observed that downregulation of ZSCAN4 in CTCF-depleted cells impaired expression of 2C markers and abrogated reprogramming to 2C-like cells (Fig. 5b, c and Supplementary Fig. 11b). Finally, due to the role of ZSCAN4 in re-activating early embryonic genes and promoting MERVL expression[37,38], we examined whether over-expression of ZSCAN4C cooperated with CTCF depletion in promoting 2C-like conversion. Indeed, although ZSCAN4C expression increased the percentage of RFP+ parental and untreated ESC[CTCF-AID] cells to a similar extent, over-expression of ZSCAN4C in CTCF-depleted ESC further boosted 2C-like conversion as early as 24 h (Fig. 5d and Supplementary Fig. 11c). These combined results demonstrated that ZSCAN4 proteins are essential for the 2C-like conversion mediated by CTCF depletion.

## Discussion

Our study demonstrates that 2C-like ESC are unstable in vitro. We observed increased DNA damage and cell death in endogenous, DUX-induced and CTCF-depleted 2C-like ESC. Similarly, over-expression of DUX in vivo leads to developmental arrest and embryo death[39]. We show that the DNA damage observed in DUX-induced 2C-like ESC is, at least partially, associated with RS mediated by DUX-induced transcription and involves the generation of single or double strand breaks at certain CTCF sites. We propose that DUX-induced transcriptional activity of otherwise silenced genes might induce local de novo transcription/replication conflicts promoting fork stalling and eventual breakage in proximal CTCF binding sites. Importantly, END-seq is a very sensitive technique that allowed us to map precisely DNA lesions recurrently happening in the same genomic location. Thus, it is likely that CTCF-associated DNA damage represents only a fraction of the total DNA damage generated in DUX-expressing ESC. Indeed, non-recurrent random breaks will be indistinguishable from background in END-seq experiments. In support of this idea, additional sources of damage have been associated with the 2C-like state or induced by DUX. In fact, human ortholog DUX4 mediates the accumulation of dsRNA foci and the activation of the dsRNA response contributing to the apoptotic phenotype associated with DUX over-expression[40]. Further work will be needed to understand the exact origin of these DNA breaks and whether the single DNA ends detected are precursor lesions to double strand brakes or are generated due to specific replication or transcriptional mechanisms.

CTCF depletion triggers spontaneous 2C-like conversion and promotes the acquisition of 2C-like features in ESC (Fig. 3). In

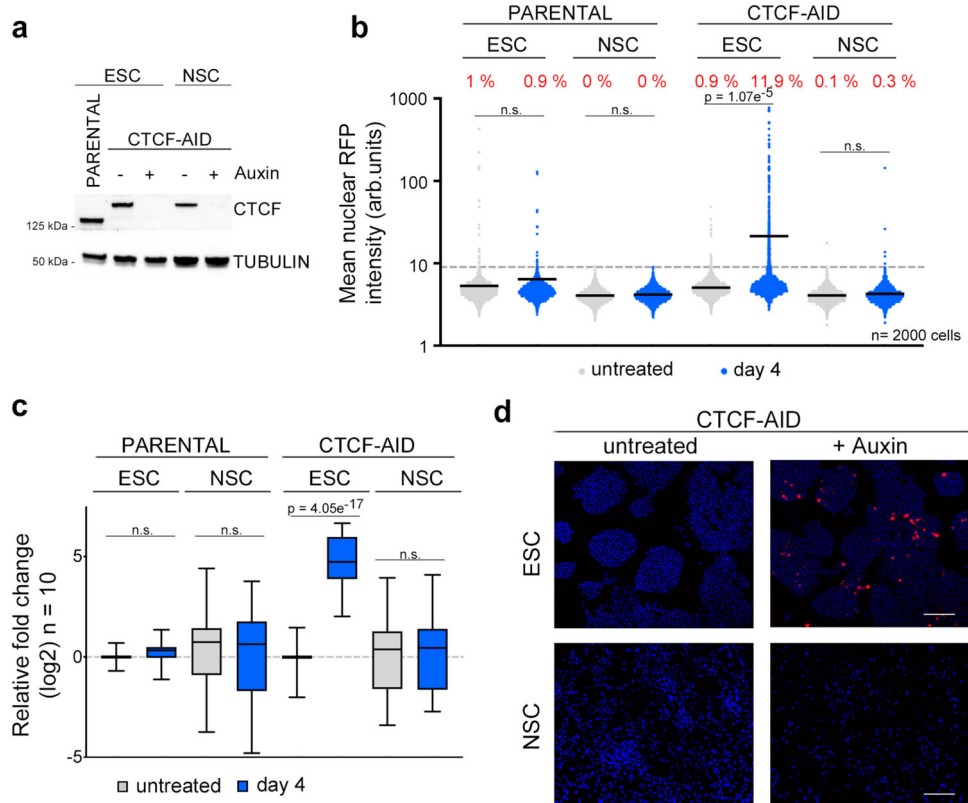

**Fig. 4 Epigenetic and/or transcriptional roadblocks prevent 2C-like conversion in CTCF-depleted ESC[CTCF-AID].** **a** Western blot analysis of CTCF performed in ESC[CTCF-AID] and NSC[CTCF-AID] untreated or treated with auxin for four days. Parental ESC were used to show the smaller size of wild-type CTCF compared to tagged CTCF-AID. Tubulin levels are shown as a loading control. **b** HTI quantification of RFP+ cells in untreated or auxin-treated for four days LTR-RFP reporter parental ESC, NSC, ESC[CTCF-AID] and NSC[CTCF-AID]. Center lines indicate mean values. Percentages of RFP+ cells above the threshold are indicated. $n = 2000$; p-values are shown from two-tailed unpaired $t$-tests. n.s. = non-significant. Three independent experiments were performed but one representative is shown. **c**, Box and whisker plot showing the relative fold change (log2) expression of a subset of ten 2C associated genes (DUX, ZSCAN4, ZFP352, TCSTV3, SP110, TDPOZ1, DUB1, EIF1ad8, PRAMEL7 and MERVLs) in ESC, NSC, ESC[CTCF-AID] and NSC[CTCF-AID] untreated or treated with auxin for four days. Reactions were performed by triplicate in two independent experiments. Center line indicates the median, box extends from the 25th to 75th percentiles and whiskers show Min to Max values. p values are shown from two-tailed paired $t$-tests. n.s. = non-significant. **d** Representative images from ESC[CTCF-AID] and NSC[CTCF-AID] untreated or treated with auxin for four days showing the presence of RFP+ cells. Scale bar, 200 μm. DAPI was used to visualize nuclei. In **a**–**d**, one representative experiment is shown but two independent experiments with similar results were performed using two different ESC clones.

addition, we showed that expression of the ZSCAN4 gene cluster is a necessary early event to successfully promote 2C-like reprogramming upon CTCF-depletion. Similarly, ZSCAN4 downregulation compromises proper embryo development and efficient somatic cell nuclear transfer performed with cohesin-depleted somatic nuclei[23,29]. Nevertheless, the precise role of the ZSCAN4 cluster in the 2C-like reprogramming is unclear. The known role of ZSCAN4 in the re-activation of early embryonic genes and promotion of MERVL expression[37,38], suggest that might be essential for successful 2C-like conversion. Moreover, ZSCAN4 has also been implicated in the maintenance of telomeres and genome stability of ESCs as well as in protecting the 2C embryo from DNA damage[41–43]. Thus, ZSCAN4 could participate in limiting the damage associated with the 2C-like conversion. Expression of DUX, which is a later event likely induced by secondary events and not by the direct loss of CTCF, enhances the transcriptional activation of the ZSCAN4 cluster by direct DUX binding to its promoters. In fact, DUX knockout ESC and embryos showed defective ZSCAN4 activation[9,10]. This positive feedback loop might be required to promote the 2C-like state.

Transition from totipotency to pluripotency during embryonic development is characterized by the progressive accumulation of CTCF and maturation of TADs[24–26]. Interestingly, CTCF binds

to a large number of endogenous RNAs and this interaction seems important for chromatin CTCF deposition[44]. Indeed, CTCF mutants unable to bind RNA showed decreased genome-wide binding[44]. It is tempting to speculate that the progressive strength of TADs during ED[24–26] correlates with increasing levels of CTCF and RNA transcription after ZGA. Further work will be needed to address how CTCF deposition and TAD insulation take place during early development and if these events play an active role in promoting the exit from totipotency in the early embryo.

Over the past decade, multiple studies have demonstrated that lineage commitment and cell identity are actively reinforced to resist cell fate changes[45]. The best example of these studies is the somatic cell reprogramming into induced pluripotent stem cells (iPSC), which is a very inefficient process. The low reprogramming efficiency is explained by epigenetic roadblocks that need to be overcome to undergo successful reprogramming[45]. Our data also suggest that the 2C-like reprogramming mediated by CTCF-depletion has to overcome similar roadblocks explaining the incomplete reprogramming in every individual CTCF-depleted pluripotent cell.

Finally, the fact that CTCF depletion leads to the reactivation of the 2C transcriptional program is of relevance if we consider

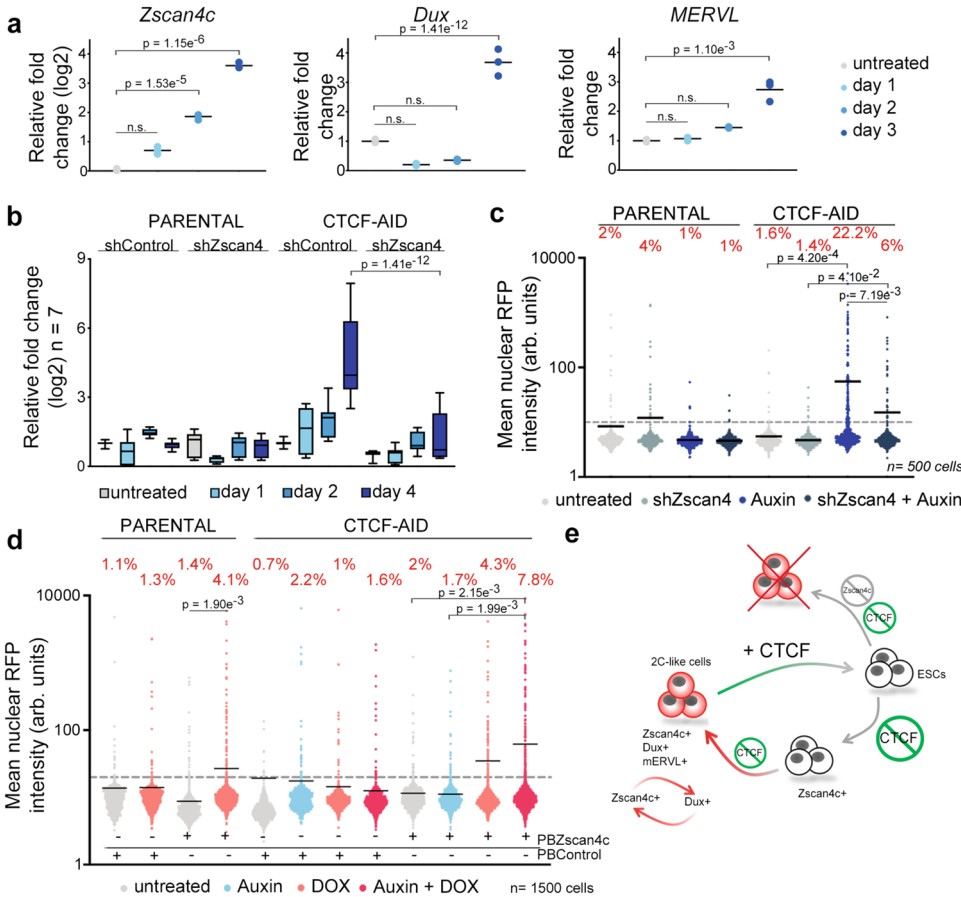

**Fig. 5 Transcriptional activation of the ZSCAN4 cluster is required for 2C-like reprogramming. a** Graph showing the relative fold change (log2 or linear) expression of DUX, ZSCAN4 and MERVL in untreated or auxin-treated LTR-RFP reporter ESC^CTCF-AID for the indicated days. Center line shows the mean. Data are shown by triplicate. *p*-values are shown from two-tailed unpaired *t*-tests. n.s. = non-significant. **b** Box and whisker plot showing the averaged relative fold change (log2) expression of seven 2C-associated genes (DUX, ZSCAN4, ZFP352, TCSTV3, SP110, TDPOZ1 and MERVLs) in untreated or auxin-treated at the indicated time points in LTR-RFP reporter control and ESC^CTCF-AID. ESC were infected with lentiviruses expressing control or shRNAs against ZSCAN4. Reactions were performed by triplicate in two independent experiments. Center line indicates the median, box extends from the 25^th to 75^th percentiles and whiskers show Min to Max values. *p*-values are shown from two-tailed paired *t*-tests. Group comparisons at Day 1 and Day 2 in ESC^CTCF-AID are non-significant. All comparisons in parental cells are non-significant. **c** HTI quantification of RFP^+ cells in untreated or auxin-treated for four days LTR-RFP reporter control and ESC^CTCF-AID. ESC were infected with lentiviruses expressing control or shRNAs against ZSCAN4. Center lines indicate mean values. *n* = 500; *p*-values are shown from two-tailed unpaired *t*-test. All comparisons in parental cells are non-significant. **d** HTI quantification of RFP^+ cells in untreated or auxin-treated for 24 h LTR-RFP reporter control and ESC^CTCF-AID. ESC harbor a DOX-inducible piggyBac (PB) construct expressing ZSCAN4C and were induced with 1 μg/mL DOX as indicated together with auxin for 24 h. Center lines indicate mean values. *n* = 1500; *p*-values are shown from two-tailed unpaired *t*-tests. **e** Schematic representation of the model inferred from the data presented here. In **a**–**d**, one representative experiment is shown but two independent experiments were performed with similar results. In **c**, **d** percentages of RFP^+ cells above the threshold are indicated.

the possibility that somatic cells with compromised CTCF functionality could re-express genes from this program. Indeed, *Ctcf* hemizygous mice are prone to spontaneous and induced cancer development in many tissues demonstrating that CTCF is haploinsufficient for tumor suppression[46,47]. In agreement, somatic missense and non-sense CTCF mutations have been commonly found in human cancers[48,49]. Interestingly, it has recently been recognized that a broad range of human cancers are characterized by an early zygotic gene signature[50]. Additional studies will be needed to determine whether CTCF functionality is altered in this subtype of human cancers.

In summary, our work revealed the important intertwined relationship between CTCF and 2C-associated features.

## Methods

**Embryo culture.** C57BL/6J mice were obtained from the Jackson Laboratory. All the animal work included here was performed in compliance with the NIH Animal Care & Use Committee (ACUC) Guideline for Breeding and Weaning. Mice were maintained in a dark/light cycle of 12 h each in a temperature range of 68°–76°F and a range of 30–70% humidity. For embryo isolation, 4-weeks old female mice were injected intraperitoneally with 5IU Pregnant Mare Serum Gonadotropin (PMSG, Prospec) followed by 5 IU human Chorionic Gonadotropin (hCG, Sigma-Aldrich) 46-48 h later. Pregnant females were euthanized, and embryos collected in M2 media (MR-015-D, Sigma-Aldrich) at indicated time points after hCG injection: E0.5, E1.0, E2.5 and E3.5. The sex of embryos was not determined. Isolated embryos were fixed for 10 min in 4% Paraformaldehyde (Electron Microscopy Sciences), permeabilized for 30 min in 0.3% Triton X-100 and 0.1 M Glycine in PBS 1X and blocked for 1 h (1% BSA, 0.1% Tween in PBS 1X), followed by overnight incubation with primary antibodies against CTCF (1:1000 dilution, ab188408, Abcam). Embryos were washed in 0.1% Tween in PBS 1X and incubated with the appropriate secondary antibody for 1 h at room temperature. Embryos were imaged using a Nikon Ti2-E microscope (Nikon Instruments) equipped with a Yokogawa CSU-W1 spinning disk unit, a Photometrics BSI sCMOS camera and 20x (N.A. 0.75) and 60x (N.A. 1.49) plan-apochromat objective lenses. Confocal z-stacks were acquired and used to generate 3D surfaces were rendered based on nuclear DAPI-staining and the corresponding regions were used to quantify the fluorescence intensity of CTCF. Embryo z-stack images were quantified using Imaris Bitplane (Oxford Instruments).

**Cell culture**. Wild-type (R1, G4, E14) ESC, ESC[DUX] and ESC[CTCF-AID] (ID: EN52.9.1 and EN204.3)[27] were grown on a feeder layer of growth-arrested MEFs or on gelatin 0.1% in high-glucose DMEM (Gibco) supplemented with 15% FBS, 1:500 LIF (made in house), 0.1 mM nonessential amino acids, 1% glutamax, 1 mM Sodium Pyruvate, 55 mM β-mercaptoethanol, and 1% penicillin/streptomycin (all from ThermoFisher Scientific) at 37 °C and 5% CO$_2$. Cells were routinely passaged with Trypsin 0.05% (Gibco). Media was changed every other day and passaged every 2-3 days. HEK293T (American Type Culture Collection) cells were grown in DMEM, 10% FBS, and 1% penicillin/streptomycin. Generation of infective lentiviral particles and ESC infections were performed as described[51]. All experiments were performed using both ESC[CTCF-AID] (ID: EN52.9.1 and EN204.3)[27] with similar results. However, only experiments using ESC[CTCF-AID] (ID: EN52.9.1) are shown throughout the manuscript.

To generate ESC[DUX] cell lines, a FLAG-tag version of the codon-optimized mouse DUX was amplified by PCR (Primers in Supplementary Table 1) from pCW57.1-mDUX-CA (Gift from Steven Tapscott, Addgene 99284) and subcloned into the pBS31 plasmid (pBS31-FLAG_mDUX). A Flp-dependent recombination event using pBS31-FLAG_mDUX in the KH2 ESC line was used to knock-in the cDNA for FLAG_mDUX into a tetO-minimal promoter allocated in the Col1a1 locus as described[11].

To generate additional ESC[CTCF-AID] cell lines, R1 and ESC[DUX] were co-transfected using jetPRIME (PolyPlus transfection) with the plasmids CTCF-AID [71-114]-eGFP-FRT-Blast-FRT (Gift from Benoit Bruneau, 92140, Addgene), pCAGGS-Tir1-V5-BpA-Frt-PGK-EM7-NeoR-bpA-Frt-Rosa26 (Gift from Benoit Bruneau, 86233, Addgene) and the plasmid pX330-U6-Chimeric_BB-CBh-hSpCas9 (Gift from Feng Zhang, 42230, Addgene) encoding sgRNAs targeting CTCF and ROSA26 alleles (see Supplementary Table 1 for sgRNA sequences). Two days after transfection ESC were selected with Neomycin (200 μg/mL) for one additional week. Individual ESC clones were picked and amplified based on eGFP expression indicating successful CTCF targeting. HTI and western blot analyses were used to verify that GFP and CTCF were lost upon addition of 500 μM auxin for 24 h.

To generate ESC lines carrying the LTR-RFP reporter, the LTR sequence was PCR amplified and subcloned in a piggyBac plasmid upstream of a turboRFP (RFP) coding region to generate the LTR-RFP reporter (Primers in Supplementary Table 1). PiggyBac-LTR-RFP plasmid together with a plasmid encoding for a supertransposase were co-transfected in ESC and further selected with Neomycin (200 μg/mL) for one week. To generate ESC[CTCF-AID] lines carrying a DOX-inducible ZSCAN4-PiggyBac construct, the coding sequence for ZSCAN4C was amplified from cDNA and subcloned into the plasmid PB-TRE-dCas9-VPR (Gift from George Church, 63800, Addgene), after removing the dCas9-VPR insert. DOX-inducible PiggyBac-ZSCAN4C plasmid together with a plasmid encoding for a supertransposase were co-transfected in ESC and further selected with Hygromycin (200 μg/mL) for one week. To generate ZSCAN4-knockdown ESC[CTCF-AID] lines, cells were infected with pLKO.1 control or pLKO.1-shZSCAN4 (5′-GAATGCAACAACTCTTGTAATCTCGAGATTACAAGAGTTGTTGCATT CT-3′, Millipore Sigma) and further selected with Puromycin (1 μg/mL) for one week. This shRNA has a perfect sequence match with the isoforms ZSCAN4C, D and F, one mismatch with ZSCAN4A and two mismatches with ZSCAN4B.

To induce differentiation of ESC[CTCF-AID] (ID: EN52.9.1 and EN204.3)[27] toward neural progenitor cells (NPCs) we seeded 0.5×10$^6$ ESC[CTCF-AID] in a 10 cm plate. The following day, media was changed to N2/B27 medium: DMEM/F12 and Neurobasal (1:1), N2 supplement, B27 supplement, 1% glutamax, 55 mM β-mercaptoethanol, and 1% penicillin/streptomycin (all from ThermoFisher Scientific) and refreshed daily for a total of 7 days. On day 7, cells were dissociated with TryplE (Gibco) and 3×10$^6$ cells were plated in low-binding plates in N2B27 with 10 ng/mL EGF (PeproTech) and FGF (R&D Systems) to promote the growth in suspension as spheres. Three days later, cell aggregates were plated in gelatinized plates and grown as a monolayer of NSCs in N2B27 with 10 ng/mL EGF/FGF. After 2-4 days, cells were passaged at least five times with Accutase before performing experiments. To avoid contamination with NSCs where the Tir1 transgene gets silenced, we pulsed NSC[CTCF-AID] with Auxin for 24 h and sorted the GFP⁻ cells. We performed the same experiments on NSC[CTCF-AID] derived from both ESC[CTCF-AID] (ID: EN52.9.1 and EN204.3)[27] with similar results. However, only one set of NSC[CTCF-AID] (ID: EN52.9.1) is shown in Fig. 4 and Extended Fig. 9.

**Immunofluorescence**. Cells were fixed in 4% Paraformaldehyde (PFA, Electron Microscopy Sciences) for 10 min at RT followed by 10 min of permeabilization using the following permeabilization buffer (100 mM Tris-HCl pH 7.4, 50 mM EDTA pH 8.0, 0.5 % Triton X-100). The following primary antibodies were incubated overnight: OCT3/4 (1:100, sc-5279, Santa Cruz Biotechnology), ZSCAN4 (1:2000, AB4340, Millipore Sigma), γH2AX (1:1000, 05-636, Millipore), CTCF (1:1000, ab188408, Abcam), Flag (1:500, F1804, Sigma Aldrich). Corresponding Alexa Fluor 488 Chicken anti-Rabbit IgG (H+L) (Thermo Fisher Scientific, Cat# 31431), Alexa Fluor 488 Goat anti-Mouse IgG (H+L) (Thermo Fisher Scientific, Cat# A-11001), Alexa Fluor 647 Chicken anti-Rabbit IgG (H+L) (Thermo Fisher Scientific, Cat# A-21443) or Alexa Fluor 647 Chicken anti-Mouse IgG (H+L) (Thermo Fisher Scientific, Cat# A-21463) secondary antibodies were used to reveal primary antibody binding (1:1000). For generating the plots shown in Extended Data 2b, image analysis was performed using a custom Python script. In brief,

DAPI-stained nuclei were segmented using the StarDist deep-learning image segmentation[52]. Segmented nuclei ROIs were used to quantify total DAPI intensity and RFP mean intensity.

**High throughput imaging (HTI)**. A total of 10,000-20,000 ESC (depending on the experiment and on the specific ESC line) were plated on gelatinized μCLEAR bottom 96-well plates (Greiner Bio-One, 655087). ESC were treated with DOX (different concentrations in the range from 150–600 ng/mL) or 500 μM auxin as indicated or incubate with 10μM EdU (Click Chemistry Tools) for 30 min before fixation with 4% PFA in PBS for 10 min at room temperature. γH2AX and ZSCAN4 staining was performed using standard procedures. EdU incorporation was visualized using Alexa Fluor 488-azide or Alexa Fluor 647-azide (Click Chemistry Tools) Click-iT labeling chemistry and DNA was stained using DAPI (4′,6-diamidino-2-phenylindole). When indicated, ESC[DUX] were treated with 1 μM ATR inhibitor (AZ20, Selleckchem).

Cooperation between CTCF-depletion and DUX expression was examined in CTCF-AID targeted ESC[DUX] upon treatment with auxin and low concentration of DOX. Similarly, cooperation between CTCF-depletion and HDAC inhibition was examined in ESC[CTCF-AID] treated with auxin and 10 μM HDAC inhibitor.

Images were automatically acquired using a CellVoyager CV7000 high throughput spinning disk confocal microscope (Yokogawa, Japan). Each condition was performed in triplicate wells and at least 9 different fields of view (FOV) were acquired per well. High-Content Image (HCI) analysis was performed using the Columbus software (PerkinElmer). In brief, nuclei were first segmented using the DAPI channel. Mean fluorescence intensities for γH2AX, ZSCAN4, CTCF, eGFP or RFP signal were calculated over the nuclear masks in their respective channels. Single cell data obtained from the Columbus software was exported as flat tabular. txt files, and then analyzed using RStudio version 1.2.5001, and plotted using Graphpad Prism version 9.0.0.

When analyzing HTI data, we considered statistically significant those samples that when compared showed a 1.5-fold difference in the averaged mean and a unpaired two-tail T-test with a p-value of at least 0.05 or lower.

**Live cell imaging**. When indicated, ESC were infected with a lentiviral plasmid encoding H2B-GFP (kind gift from Marcos Malumbres, CNIO, Spain). A total of 40,000 ESC were plated in gelatin-coated μ-Slide 8 wells plates (80826, Ibidi) and imaged untreated or Auxin/DOX-treated for a time period between 43–48 h depending on the experiment. Images were acquired every 15 or 20 min over the time course using either a Nikon spinning disk confocal microscope or a Zeiss LSM780 confocal microscope equipped with 20x plan-apochromat objective lenses (N.A. 0.75 and 0.8, respectively) and stage top incubators to maintain temperature, humidity and CO$_2$ (Tokai Hit STX and Okolab Bold Line, respectively).

**Western blot**. Trypsinized cells were lysed in 50 mM Tris pH 8, 8 M Urea (Sigma) and 1% Chaps (Millipore) followed by 30 min of shaking at 4 °C. 20 μg of supernatants were run on 4–12% NuPage Bis-Tris Gel (Invitrogen) and transferred onto Nitrocellulose Blotting Membrane (GE Healthcare). Membranes were incubated with the following primary antibodies overnight at 4 °C: p-KAP1 (dilution 1:1000, A300-767A, Bethyl) or ZSCAN4C (1:500, AB4340, Millipore Sigma), γH2AX (1:1000, 05-636, Millipore), CTCF (1:1000, 07-729, Millipore), Flag (1:1000, F1804, Sigma Aldrich), Tubulin (1:50000, T9026, Sigma-Aldrich). The next day the membranes were incubated with HRP-conjugated secondary antibodies Goat anti-Rabbit IgG (H+L) (1:5000; Thermo Fisher Scientific, Cat# 31466) or Goat anti-Mouse IgG (H+L) (1:5000; Thermo Fisher Scientific, Cat# 31431) for 1 h at room temperature. Membranes were developed using SuperSignal West Pico PLUS (Thermo Scientific).

**Flow cytometry and cell sorting**. For live cell flow cytometry experiments, cells were dissociated into single cell suspensions and analyzed for RFP expression, DAPI was added to detect cells with compromised membrane integrity. For EdU Click-IT experiments, cells were incubated for 20 min with 10 μM EdU, fixed in 4% paraformaldehyde, permeabilized in 0.5% triton X-100, followed by Alexa Flour 488-azide or Alexa Flour 647-azide Click-iT labeling chemistry. DNA content was stained using DAPI or Hoechst 33342 (62249, Thermo Fisher Scientific). Analytic flow profiles were recorded on a LSRFortessa (BD Biosciences) or a FACSymphony A5 instrument (BD Biosciences). Data was analyzed using FlowJo Version 10.7.1. Cell sorting experiments were performed on a BD FACSAria Fusion instrument. Post-sort quality control was performed for each sample.

**RNA extraction, cDNA synthesis and qPCR**. Total RNA was isolated using Isolate II RNA Mini Kit (Bioline). cDNA was synthesized using SensiFAST cDNA Synthesis Kit (Bioline). Quantitative real time PCR was performed using iTaq Universal SYBR Green Supermix (BioRad) in a CFX96 Touch BioRad system. Expression levels were normalized to GAPDH. For a primer list see Supplementary Table 1. When analyzing quantitative real time PCR data, we considered statistically significant those samples that when compared showed an averaged of two-fold difference in overall gene expression and an unpaired two-tail T-test with a p-value of at least 0.05 or lower.

**CUT&RUN protocol**. The CUT&RUN protocol was slightly modified as described[53,54]. In brief, trypsinized or cell sorted ESC (between 150,000–500,000 cells depending on the experiment) were washed three times with Wash Buffer (20 mM HEPES-KOH pH 7.5, 150 mM NaCl, 0.5 mM spermidine, Roche complete Protease Inhibitor tablet EDTA free) and bound to activated Concanavalin A beads (Polysciences) for 10 min at room temperature. Cells were then permeabilized in Digitonin Buffer (0.05% Digitonin and 0.1% BSA in Wash Buffer) and incubated with 4 μL of the antibody against CTCF (07-729, Millipore) at 4 °C for 2 h. For negative controls, Guinea Pig anti-Rabbit IgG (ABIN101961, Antibodies-online) was used. Cells were washed with Digitonin Buffer following antibody incubation, and further incubated with purified hybrid protein A-protein G-Micrococcal nuclease (pAG-MNase) at 4 °C for 1 h. Samples were washed in Digitonin Buffer, resuspended in 150 μL Digitonin Buffer and equilibrated to 0 °C on ice water for 5 min. To initiate MNase cleavage, 3 μL 100 mM CaCl$_2$ was added to cells and after 1 h of digestion, reactions were stopped with the addition of 150 μL 2x Stop Buffer (340 mM NaCl, 20 mM EDTA, 4 mM EGTA, 0.02% Digitonin, 50 μg/mL RNase A, 50 μg/mL Glycogen). Samples were incubated at 37 °C for 10 min to release DNA fragments and centrifuged at 16,000 g for 5 min. Supernatants were collected and a mix of 1.5 μL 20% SDS/2.25 μL 20 mg/mL Proteinase K was added to each sample and incubated at 65 °C for 35 min. DNA was precipitated with ethanol and sodium acetate and pelleted by high-speed centrifugation at 4 °C, washed, air-dried and resuspended in 10 μ 0.1x TE.

**Library preparation and sequencing**. The entire precipitated DNA obtained from CUT&RUN was used to prepare Illumina compatible sequencing libraries. In brief, end-repair was performed in 50 μL of T4 ligase reaction buffer, 0.4 mM dNTPs, 3 U of T4 DNA polymerase (NEB), 9 U of T4 Polynucleotide Kinase (NEB) and 1 U of Klenow fragment (NEB) at 20 °C for 30 min. End-repair reaction was cleaned using AMPure XP beads (Beckman Coulter) and eluted in 16.5 μL of Elution Buffer (10 mM Tris-HCl pH 8.5) followed by A-tailing reaction in 20 μL of dA-Tailing reaction buffer (NEB) with 2.5 U of Klenow fragment exo- (NEB) at 37 °C for 30 min. The 20 μL of the A-tailing reaction were mixed with Quick Ligase buffer 2X (NEB), 3000 U of Quick Ligase (NEB) and 10 nM of annealed adapter (Illumina truncated adapter) in a volume of 50 μL and incubated at room temperature for 20 min. The adapter was prepared by annealing the following HPLC-purified oligos: 5′-Phos/GATCGGAAGAGCACACGTCT-3′ and 5′-ACACTCTTTCCCTA-CACGACGCTCTTCCGATC*T-3′ (*phosphorothioate bond). Ligation was stopped by adding 50 mM of EDTA, cleaned with AMPure XP beads and eluted in 14 μL of Elution Buffer. All volume was used for PCR amplification in a 50 μL reaction with 1 μM primers TruSeq barcoded primer p7, 5′-CAAGCAGAA-GACGGCATACGAGATXXXXXXXXGTGACTGGAGTTCAGACGTGTGCTCTTCCGATC*T-3′ and TruSeq barcoded primer p5 5′-AATGATACGGCGACCACCGAGATCTACACXXXXXXXXACACTCTTTCCCTACACGACGCTCTTCCGATC*T-3′ (*represents a phosphothiorate bond and XXXXXXXX a barcode index sequence), and 2X Kapa HiFi HotStart Ready mix (Kapa Biosciences). The temperature settings during the PCR amplification were 45 s at 98 °C followed by 15 cycles of 15 s at 98 °C, 30 s at 63 °C, 30 s at 72 °C and a final 5 min extension at 72 °C. PCR reactions were cleaned with AMPure XP beads (Beckman Coulter), run on a 2% agarose gel and a band of 300 bp approximately was cut and gel purified using QIAquick Gel Extraction Kit (QIAGEN). Library concentration was determined with KAPA Library Quantification Kit for Illumina Platforms (Kapa Biosystems). Sequencing was performed on the Illumina NextSeq550 (75 bp pair-end reads).

**Cut&Run data processing**. Data were processed using a modified version of Cut&RunTools[55] (Supplementary Software 1). Reads were adapter trimmed using fastp v.0.20.0[40]. An additional trimming step was performed to remove up to 6 bp adapter from each read. Next, reads were aligned to the mm10 genome using bowtie2[56] with the 'dovetail' and 'sensitive' settings enabled. Alignments were further divided into ≤ 120-bp and > 120-bp fractions. Alignments from the ≤ 120-bp fractions were downsampled to the lowest depth sample (13 million mapped ≤ 120-bp fragments for ESC$^{Dux}$ cells and 11 million ≤120-bp for spontaneously converting 2C-like ESC) before peak calling. Peaks were called using SEACR using the "stringent" peak selection mode and a corresponding IgG control[57]. Normalized (RPKM) signal tracks were generated using the 'bamCoverage' utility from deepTools with parameters bin-size = 25, smooth length = 75, and 'center_reads' and 'extend_reads' options enabled[58].

**RNAseq data processing and batch correction**. Fastq files for RNAseq experiments[5,27] were downloaded from SRA. RNAseq reads were adapter trimmed using fastp v.0.20.0[59]. Transcript expression was quantified via mapping to mouse gencode v25 transcripts using salmon[60]. In order to compare the two RNAseq experiments, batch correction was performed. Gene counts across samples were quantile-normalized using the limma package[61]. Batch correction was then performed on quantile-normalized counts using COMBAT[62].

**END-seq**. END-seq was performed as described[63]. Briefly, for untreated DOX-treated ESC$^{DUX}$, a total of 30 million cells in single cell suspension were embedded in a single agarose plug. Lysis and digestion of embedded cells was performed using

Proteinase K (50 °C, 1 h then 37 °C for 7 h). Agarose plugs were rinsed in TE buffer and treated with RNase A at 37 °C, 1 h. Next, DNA ends were blunted. For these reactions, DNA was retained in the plugs to prevent shearing. The first blunting reaction was performed using ExoVII (NEB, M0379S) for 1 hr, 37 C. Plugs were washed twice in NEB Buffer 4 (1X), immediately followed by the second blunting reaction using ExoT (NEB, M0265S) for 1 h, 24 °C. After this final blunting, two washes were performed in NEBNext dA-Tailing Reaction Buffer (NEB, B6059S), followed by A-tailing (Klenow 3′- > 5′ exo-, NEB, M0212S). After A-tailing, we performed a ligation with the "END-seq hairpin adapter 1," listed in reagents section, using NEB Quick Ligation Kit (NEB, M2200S).

**DNA sonication, end-repair, A-tailing, and library amplification**. Agarose plugs were then melted and dissolved. DNA was sonicated using to a median shear length of 170 bp using a Covaris S220 sonicator for 4 min at 10% duty cycle, peak incident power 175, 200 cycles per burst, 4 °C. Following the sonication, DNA was precipitated with ethanol and dissolved in 70 μL TE buffer. 35 μL of Dynabeads were washed twice with 1 mL Binding and Wash Buffer (1xBWB) (10 mM Tris-HCl pH8.0, 1 mM EDTA, 1 M NaCl, 0.1% Tween20). After the wash, beads were recovered using a DynaMag-2 magnetic separator (12321D, Invitrogen) and supernatants were discarded. Washed beads were resuspended in 130 μL 2xBWB (10 mM Tris-HCl pH8.0, 2 mM EDTA, 2 M NaCl) combined with the 130 μL of sonicated DNA followed by an incubation at 24 °C for 30 min. Next, the supernatant was removed, and the biotinylated DNA bound to the beads was washed thrice with 1 mL 1xBWB, twice with 1 mL EB buffer, once with 1 mL T4 ligase reaction buffer (NEB) and then resuspended in 50 μL of end-repair reaction mix (0.4 mM of dNTPs, 2.7 U of T4 DNA polymerase (NEB), 9 U of T4 Polynucleotide Kinase (NEB) and 1 U of Klenow fragment (NEB)) and incubated at 24 °C for 30 min. Once again, the supernatant was removed using a magnetic separator and beads were then washed once with 1 mL 1xBWB, twice with 1 mL EB buffer, once with 1 mL NEBNext dA-Tailing reaction buffer (NEB) and then resuspended in 50 μL of with NEBNext dA-Tailing reaction buffer (NEB) and 20 U of Klenow fragment exo- (NEB). The A-tailing reaction was incubated at 37 °C for 30 min. The supernatant was removed using a magnetic separator and washed once with 1 mL NEBuffer 2 and resuspended in 115 mL of Ligation reaction with Quick Ligase buffer (NEB), 6,000 U of Quick Ligase (NEB) and ligated to "END-seq hairpin adapter 2" by incubating the reaction at 25 °C for 30 min. Reaction was stopped by adding 50 mM of EDTA, and beads washed 3X BWB, 3X EB, and eluted in 8 μL of EB. Hairpin adapters were digested using USER enzyme (NEB, M5505S) at 37 C for 30 min. PCR amplification was performed in 50 μL reaction with 10 mM primers 5′-CAAGCAGAAGACGGCATACGA-GATXXXXXXXGTGACTGGAGT TCAGACGTGTGCTCTTCCGATC*T-3′ and 5′-AATGATACGGCGACCACC GAGATCTACACTCTTTCCCTACACGACGCTCTTCCGATC*T-3′, and 2X Kapa HiFi HotStart Ready mix (Kapa Biosciences). * represents a phosphothioratebond and NNNNNN a Truseq index sequence. PCR program: 98 °C, 45 s; 15 cycles [98 °C, 15 s; 63 °C, 30 s; 72 °C, 30 s]; 72 °C, 5 min. PCR reactions were cleaned with AMPure XP beads, and after running the reactions on a 2% agarose gel, 200-500 bp fragments were isolated. Libraries were purified using QIA-quick Gel Extraction Kit (QIAGEN). Library concentration was determined with KAPA Library Quantification Kit for Illumina Platforms (Kapa Biosystems) and the sequencing was performed on Illumina NextSeq 500 or 550 (75 bp single end reads).

**Processing of END-seq data**. END-seq reads were aligned to the mouse reference genome mm10 using bowtie (v1.1.2)[56] (PMID: 19261174) with parameters -n 3 -l 50 -k 1. Functions "view" and "sort" of samtools (v 1.6) (PMID: 19505943) were used to convert and sort the aligned sam files to sorted bam files. Bam files were further converted to bed files by bedtools bamToBed command (PMID: 20110278). END-seq peaks were called by MACS (v1.4.3)[64] with parameters–nolambda–nomodel–keep-dup=all (PMID: 18798982) and peaks within blacklisted regions (https://sites.google.com/site/anshulkundaje/projects/blacklists) were filtered out (PMID: 31249361). Overlapped peaks from two independent clones were used in this paper. To determine whether the END-seq peak corresponded to a single or double strand lesion (asymmetric versus symmetric) we calculated the ratio between the signal intensity per strand. If this value was found to be between −1 and 1 the lesion was considered symmetric or double stranded. If the ratio was higher than 1 or lower than −1 the lesion was considered asymmetric or single stranded. Gene association was performed by using GREAT (http://great.stanford.edu/public/html/) using "single nearest gene" by default 1000 kb distance.

**ChIP-seq**. Twenty million of untreated or DOX-treated ESC$^{DUX}$ cells were fixed using 1% Formaldehyde (Sigma, F1635) at 37 °C for 10 min. Fixation was then quenched with 125 mM glycine (Sigma). Cell pellets were washed twice with cold PBS and samples were snap-frozen and stored in −80 °C. Frozen pellets were resuspended in 1 mL RIPA buffer (10 mM Tris-HCl pH 7.5, 1 mM ethylenedia-minetetraacetic acid (EDTA), 0.1% sodium dodecyl sulfate, 0.1% sodium deoxycholate, 1% Triton X-100, and 1 Complete Mini EDTA-free proteinase inhibitor tablet (Roche)). Sonication was performed using Covaris S220 (duty cycle 20%, peak incident power 175, and cycle/burst 200 for 30 min at 4 °C). Chromatin was pre-clarified with 40 μL prewashed Dynabeads Protein A (ThermoFisher) for

30 min at 4 ℃ and then incubated with 40 µL Dynabeads Protein A bound to 10µg of anti-RPA32 antibody (Abcam ab10359) or 10µg of Guinea Pig anti-Rabbit IgG (ABIN101961, Antibodies-online) in 100 µL PBS overnight at 4 ℃. Beads were then collected in a magnetic separator (DynaMag-2 Invitrogen), washed twice with cold RIPA buffer, twice with RIPA buffer containing 0.3 M NaCl, twice with LiCl buffer (0.25 M LiCl, 0.5% Igepal-630, 0.5% sodium deoxycholate), once with TE (10 mM Tris pH 8.0, 1 mM EDTA) + 0.2% Triton X-100, and once with TE. Crosslinking was reversed by incubating the chromatin bound beads at 65 ℃ for 4 h in the presence of 0.3% SDS and 1 mg/mL of Proteinase K (Qiagen). Chromatin DNA extraction from beads and library preparation were performed as described[65].

**ChIP-seq data processing**. For all ChIP-seq data sets, reads were aligned to the mm10 genome using bowtie2[56] (paired end reads) or bwa mem (for single-end reads, in the case of the RPA ChIP-seq data) ref: https://pubmed.ncbi.nlm.nih.gov/19451168/ For paired-end data, duplicate reads were removed using MarkDuplicates from the Picard toolkit ("Picard Toolkit." 2019. Broad Institute, GitHub Repository. http://broadinstitute.github.io/picard/). For single-end data (RPA ChIP-seq), PCR duplicates were removed using the 'filterdup' command from macs2 v2.1.1[57], with the parameter–keep-dup="auto". Normalized (RPKM) signal tracks were generated bamCoverage utility from deepTools[59], using the parameters bin-size = 25, smooth length = 75, 'center_reads' and 'extend_reads'. For paired-end data, read mates were extended to the fragment size defined by the two read mates. For single-end ChIP-seq data, reads were extended to the estimated fragment length estimated by phantompeakqualtools[66]. Gene association was performed by using GREAT (http://great.stanford.edu/public/html/) using "single nearest gene" by default 1000 kb distance.

**Reporting summary**. Further information on research design is available in the Nature Research Reporting Summary linked to this article.

## Data availability

The sequencing data generated in this study have been deposited in the Gene Expression Omnibus database under accession code GSE165162. Datasets obtained from publicly available sources include GSE85624, GSE85627, GSE85185 and GSE22562 (https://www.ncbi.nlm.nih.gov/geo/query/acc.cgi?acc=GSE85624 / GSE85627 / GSE85185 and GSE22562, respectively). Additional data and/or reagents that support the findings of this study are available from the corresponding author upon reasonable request. Source data are provided with this paper.

## Code availability

Custom code used in this study is provided as Supplementary Software 1 with this paper and it is also available here: https://github.com/desireetillo/CTCF_CutAndRun21[67].

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

## Acknowledgements

We thank Bechara Saykali and Pedro Rocha for critical reading of the manuscript, and to Jacob Paiano for critical discussion. We are grateful to Christian Franke for the continuous technical support on R. We also thank Pedro Rocha, Rafael Casellas and Seol Kyoung Jung for their help on exploring HiC data. We thank Sagrario Ortega and the Transgenic Unit at CNIO for their initial help in this project. David Goldstein and the CCR Genomics Core for sequencing support and Ferenc Livak and the CCR Flow cytometry Core for experimental support. Research in S.R. laboratory is supported by the Intramural Research Program of the NIH. T.O. is supported by a postdoctoral fellowship of the Helen Hay Whitney Foundation.

## Author contributions

T.O. and S.R. conceived the study. T.O., M.V-S. designed, performed and analyzed experiments. C.N.D. and M.F. provided technical support. D.T. and P.C.F. analyzed sequencing data. G.P. provided support with high-throughput microscopy imaging. A.D.T. and M.J.K. analyzed confocal microscopy data. E.L.D. and M.M.P. provided critical reagents. N.Z. performed END-seq experiments. W.W. analyzed END-seq data. A.N. supervised END-seq experiments. R.P. performed ChIP-seq experiments. E.P.N. provided critical reagents. S.R. supervised the study and wrote the manuscript with comments and help from all authors.

## Competing interests

The authors declare no competing interests.
