## [Peer Review File · Nature Communications]

REVIEWER COMMENTS

Reviewer #1 (Remarks to the Author):

A well-written manuscript describing an interesting link between genetically induced reprogramming of ESC into 2C/totipotent-like cells, DNA damage and CTCF. While appealing, the model needs further substantiation and some of the claims need to be toned down. As well, bearing in mind that ESC-to-2C reprogramming is a lab-made phenomenon of uncertain relevance for understanding the differentiation of totipotent cells, the authors should try to be more convincing regarding the physiological implications of their observations.

1. The authors observe that treatment with replicative stress induced with an ATR inhibitor and DUX overexpression exert additive effects on DNA damage, and link this phenomenon to reprogramming. Do replicative stress or DNA damage alone (e.g. triggered by UV or H2O2) result in the induction of 2-cell stage markers such as MERVL, ZSCAN, or DUX?

2. It would be good to confirm the results of the END-seq with an alternative approach, for instance γ H2AX ChIP-seq.

3. The AID-CTCF protein is expressed at significantly lower level than its wild type counterpart. Are ESC homozygous for this allele fully functional, whether for differentiation or for absence of spontaneous 2C-like reprogramming, in spite of reduced CTCF levels? On Fig 3a blot, ZSCAN4 levels seem slightly higher in these than in parental cells (moreover, it is a detail, but there is also a background band that differs between the two lines).

4. Is DNA damage induced upon AID-CTCF degradation?

5. Does CTCF bind close to and directly regulates DUX4?

6. Can the authors propose an explanation for the systematically low fraction of cells converting to a 2C-like stage following their various manipulations (e.g. less than 30% upon CTCF degradation, which must presumably occur in all cells)? What is intrinsically different between ZSCAN4- or MERVL-RFP-positive and -negative cells in this experiment?

6. Statistics should be added to plots lacking them (a majority).

7. In their model DUX-> de novo transcription/replication conflict -> replicative stress / DNA damage -> 2C conversion, the authors do not explain why it would happen only with DUX and not with other transcriptional activators. As well, they find ZSCAN4 repression to prevent 2C conversion of CTCF-depleted ESC to 2C-like cells, and argue in their discussion that ZSCAN4 could participate in limiting the DNA damage associated with the conversion. What is the link between their observation and this hypothesis?

Minor comments:

1. ZSCAN4 is described as one gene all along the paper whereas it is a cluster, with some isoforms DUX4-regulated. This should be clarified.

2. Line 130: ED should be defined once (we suppose the authors mean Embryonic Development).

3. ENDseq or END-seq? Either one should be chosen, and then stuck to.

A well-written manuscript describing an interesting link between genetically induced reprogramming of ESC into 2C/totipotent-like cells, DNA damage and CTCF. While appealing, the model needs further substantiation and some of the claims need to be toned down. As well, bearing in mind that ESC-to-2C reprogramming is a lab-made phenomenon of uncertain relevance for understanding the differentiation of totipotent cells, the authors should try to be more convincing regarding the physiological implications of their observations.

1. The authors observe that treatment with replicative stress induced with an ATR inhibitor and DUX overexpression exert additive effects on DNA damage, and link this phenomenon to reprogramming. Do replicative stress or DNA damage alone (e.g. triggered by UV or H2O2) result in the induction of 2-cell stage markers such as MERVL, ZSCAN, or DUX?

2. It would be good to confirm the results of the END-seq with an alternative approach, for instance γ H2AX ChIP-seq.
3. The AID-CTCF protein is expressed at significantly lower level than its wild type counterpart. Are ESC homozygous for this allele fully functional, whether for differentiation or for absence of spontaneous 2C-like reprogramming, in spite of reduced CTCF levels? On Fig 3a blot, ZSCAN4 levels seem slightly higher in these than in parental cells (moreover, it is a detail, but there is also a background band that differs between the two lines).
4. Is DNA damage induced upon AID-CTCF degradation?
5. Does CTCF bind close to and directly regulates DUX4?
6. Can the authors propose an explanation for the systematically low fraction of cells converting to a 2C-like stage following their various manipulations (e.g. less than 30% upon CTCF degradation, which must presumably occur in all cells)? What is intrinsically different between ZSCAN4- or MERVL-RFP-positive and -negative cells in this experiment?
6. Statistics should be added to plots lacking them (a majority).
7. In their model DUX-> de novo transcription/replication conflict -> replicative stress / DNA damage -> 2C conversion, the authors do not explain why it would happen only with DUX and not with other transcriptional activators. As well, they find ZSCAN4 repression to prevent 2C conversion of CTCF-depleted ESC to 2C-like cells, and argue in their discussion that ZSCAN4 could participate in limiting the DNA damage associated with the conversion. What is the link between their observation and this hypothesis?

Minor comments:

1. ZSCAN4 is described as one gene all along the paper whereas it is a cluster, with some isoforms DUX4-regulated. This should be clarified.
2. Line 130: ED should be defined once (we suppose the authors mean Embryonic Development).
3. ENDseq or END-seq? Either one should be chosen, and then stuck to.

Reviewer #2 (Remarks to the Author):

Olbrich et al examine events following induction of a 2C-like fate from ESCs, which eventually leads to identification of a role for CTCF as a barrier to induction of the 2C state. The authors find 2C induction increases DNA damage, and the sites of damage frequently overlap with binding sites of CTCF and cohesin. CTCF binding is reduced in 2C cells relative to ESCs. Interestingly, depletion of CTCF from ESCs leads to an increase in spontaneous conversion to the 2C state. Finally, the authors show that CTCF depletion leads to upregulation of ZSCAN4 at early time points, which was previously shown to precede emergence of the full 2C phenotype (i.e., MERVL and DUX expression). Overexpression of ZSCAN4C in CTCF depleted cells increased 2C conversion and knock-down of ZSCAN4 blocked reprogramming, suggesting ZSCAN4 is critical for reprogramming to 2C observed in CTCF depleted cells.

The finding that CTCF is a barrier to reprogramming to the 2C state is of considerable interest. By and large, the experiments were well designed and the conclusions well supported. I have a few concerns and several minor points that require attention.

Major comments:

1. In Fig. 2a, enrichment is shown at the peak locations specifically from Dox-induced cells. Aggregation specifically at peaks from the Dox+ cells will bias the result. If, for example, the no Dox cells also have numerous END-seq peaks, but these peaks are in different locations than the peaks

from Dox+ cells, aggregation over the no Dox peaks would yield the opposite result – that damage is higher without Dox. Peaks from all conditions (Dox+ and Dox-) should be included in any aggregate plot of this type, particularly when the plot is used as an argument that damage is higher in Dox+ cells.

2. Given the somewhat moderate changes in aggregate CTCF enrichment (Fig. 1e), it would be worth normalizing the CUT&RUN data to the contaminating E. coli reads (see Meers et al, eLife 2019), which will serve the same purpose as spike-in normalization. Such normalization can flatten differences due to technical variation, such as library quality or read depth.

3. The gene expression data in Fig. S6c suggest that CTCF loss results in partial (but incomplete) induction of the 2C transcription pattern. (Comparing DUX-expressing LTR+ cells to auxin-treated cells shows weaker induction of 2C genes in CTCF depleted cells than in DUX expressing cells.) However, a more direct comparison would be LTR+ auxin-treated cells to LTR+ DUX-expressing cells, which was not made. In addition, the other figures describing gene expression changes upon CTCF depletion, Figures 3c, 3d, and 3g, show differences in expression of 30, 6 (in browser tracks), and 10 genes, respectively. Why are variable subsets of 2C genes quantified from experiment to experiment rather than showing the same 40+ 2C-specific genes (from S6c) in 3c and 3g? It is particularly strange to select 30 genes for one plot and 10 for another plot in the same figure.

Minor points:

1. Fig. 1b: In these images, the RFP+ cells are substantially reduced at the longest time points, which is inconsistent with the quantification in 1a. If the images are not representative of the population quantified in 1a, it is not clear why they are included.

2. A more thorough discussion of the END-seq data would be helpful, including a brief mention of how single and double stranded lesions are identified, and the significance of each class of lesion to the interpretation of what type of DNA damage may be inducing these breaks.

3. The authors discuss the reduction in CTCF binding upon induction of 2C, but there did not seem to be any mention of whether novel CTCF binding sites emerge in 2C cells. It is worth reporting the number (if any) of new CTCF binding sites found in 2C cells and discussing these in the text.

4. "...2C-like reprogramming was further boosted cooperatively by expressing low levels of DUX or by incubating ESCCTCF-AID with HDAC inhibitors, known to promote 2C-like conversion..." A more thorough description (beyond this one sentence) of the alternative 2C reprogramming methods (low levels of DUX or HDACi) would be helpful.

5. "Collectively, these results demonstrated that chromatin bound CTCF prevents 2C-like conversion." It is not clear why "chromatin bound" is included in this sentence, as it is not strictly true that depletion of the "chromatin bound" fraction was shown to cause the phenotype.

6. "Furthermore, over-expression of ZSCAN4C boosted 2C-like conversion as early as 24 hours specifically in CTCF-depleted ESC while cells with normal levels of CTCF did not show major changes in the number of 2C-like cells (Fig. 4d and Extended data Fig. 8e)." This conclusion appears to be inaccurate, as overexpression of ZSCAN4C also increases 2C-like cells in the parental cells (Fig. 4d). I do not see this as a concern about the data, but the figure should be described accurately.

7. The "b" and "c" labels for S8b and S8c seem to be swapped (or alternatively their descriptions in the legend are swapped). In addition, the immunofluorescence data in S8c would be stronger if the

fraction of cells expressing ZSCAN4, LTR-RFP, or both were quantified.

8. Several details are missing from the methods and/or figure legends. For example, the amount of Dox used in each experiment and the amount of CTCF antibody used for CUT&RUN need to be reported.

Reviewer #3 (Remarks to the Author):

Olbrich et al. show a surprising role for CTCF in preventing the switch from pluripotency to totipotency. Totipotency or 2C-like state is characterized by the upregulation of endogenous retroviruses. The authors show that inducible expression of the transcription factor DUX activate genes that are characteristic of the 2C-like state. Associated with this state are DNA breaks which overlap with the position of CTCF sites. In the 2C-like cells CTCF and CTCF binding is decreased, which is consistent with CTCF levels in 2C-stage embryos. When CTCF is acutely depleted, cells also upregulate genes associated with the 2C-like state. The exact mechanism behind this remains unclear, however, the authors show that knock-down of Zscan4, which is a target of DUX mitigates the induction of the 2C-like state as a consequence of CTCF depletion.

The manuscript presents an exciting and unexpected role for CTCF in preventing reprogramming to the totipotent state. I believe the manuscript is well presented and balanced. I recommend this paper for publication. I have a few suggestions to improve the manuscript.

* The authors discuss the decrease they observe in CTCF binding in terms of a "more relaxed chromatin architecture" (p.7 line 144). However, no evidence is shown for this and this pure conjecture. I believe it is better to steer away from these types of conclusion or show actual data. This comment is further complicated by the fact that severe decrease in CTCF can have rather mild effect on chromatin looping. How CTCF would lead to more relaxed chromatin is unknown and not discussed, as such it remains an interesting correlation between relaxed chromatin and the absence of CTCF.

* Nora et al have shown that the direct effects of CTCF depletion on gene expression are likely to be genes that have CTCF bound near their promoters. Have the authors considered that CTCF loss may lead to a decrease in the expression of repressors of the Dux gene or other regulator of the 2C-like state.

* Is there anything particular to the CTCF sites that overlap with the peaks in the END-seq data? Are they loop anchors for instance?

* Why is it surprising that knock-down of Zscan4 abrogates reprogramming to the 2C-like state? Is this not exactly why the experiment was performed?

* The effects on mRNA levels following reconstitution of CTCF (wash-off) are relatively mild. The authors could consider to do a nascent RNA sequencing experiment, which may yield a stronger regression to the pluripotent transcription state.

* It was unclear whether the Zscan4 shRNAs target only Zscan4c or other copies as well. Are the other copies not upregulated as well? Please discuss this in more detail in the main text.

* The authors mention "stabilize the 2C-like state" (p.11 line 224). This seem like a counter-intuitive

statement. The 2C stage is transient, therefore there may be no regulatory loop to stabilize the 2C stage and by extension the 2C-like state.

POINT-BY-POINT RESPONSE TO THE REVIEWERS' COMMENTS

We first would like to thank the reviewers for their comments as we believe they have truly helped us to improve considerably our first version of the manuscript. Please find below a point-by-point response to all the concerns and suggestions made by the reviewers.

REVIEWER COMMENTS

Reviewer #1 (Remarks to the Author):

A well-written manuscript describing an interesting link between genetically induced reprogramming of ESC into 2C/totipotent-like cells, DNA damage and CTCF. While appealing, the model needs further substantiation and some of the claims need to be toned down. As well, bearing in mind that ESC-to-2C reprogramming is a lab-made phenomenon of uncertain relevance for understanding the differentiation of totipotent cells, the authors should try to be more convincing regarding the physiological implications of their observations.

We appreciate the reviewer's general view about our manuscript and that he/she finds it appealing. We agree on the fact that the ESC-to-2C reprogramming process is a lab-made phenomenon. However, and similar to the process of somatic cell reprogramming to iPSC, it is also a great resource and tool to explore the molecular pathways involved in the regulation of totipotency related features as identified in the embryo. In this revised version, we have toned down some of our claims and limit the use of the word "totipotency" to where it accurately could be used. Moreover, we have included new results with additional discussion to provide our view on the physiological implications of our observations.

1. The authors observe that treatment with replicative stress induced with an ATR inhibitor and DUX overexpression exert additive effects on DNA damage, and link this phenomenon to reprogramming. Do replicative stress or DNA damage alone (e.g. triggered by UV or H2O2) result in the induction of 2-cell stage markers such as MERVL, ZSCAN, or DUX?

We appreciate the reviewer's comment as we can further clarified our initial observations, provide additional experiments and include recent works as references in this regard. Indeed, it has been very recently shown that replication stress in ESC induces genes expressed in 2C embryos and 2C-like cells (Atashpaz et al, 2020). ESC treated with replication stress agents such as hydroxyurea, aphidicolin or UV-light showed an increased percentage of Zscan4+ and MERVL+ ESC (Atashpaz et al, 2020). This effect seems to be mediated by the reactivation of endogenous DUX in an ATR-dependent manner (Atashpaz et al, 2020). Furthermore, it was also shown that ESC treated with Zeocin, which induces double strand breaks, or Cisplatin, a DNA alkylating agent, led to an increase in the number of ZSCAN4+ cells, which correlated with the levels of the DNA damage marker γ H2AX (Storm et al, 2014). Finally, we have also recently reported that a muted DNA-damage response in ESC upon TRF2-deletion also correlated with 2C-conversion (Markiewicz-Potoczny et al, 2021). To confirm some of these published results, we also examined whether aphidicolin or H2O2 treatment increases the number of MERVL+ ESC in combination

with CTCF depletion (see below in Figure a). Although we did not observe any increase in MERVL+ ESC upon H₂O₂ treatment, we confirmed as reported that aphidicolin treatment increased the percentage of 2C-like ESC. However, we did not detect cooperativity with CTCF depletion (Figure a). In our revised version, we did not include these experiments, but we have included the aforementioned recent references, which were not in our first version, and discussed these results placing more into context our observations.

Figure a: High-throughput microscopy imaging quantification of MERVL+ ESC (RFP⁺ cells) in untreated or auxin-treated for a total of four days LTR-RFP reporter ESC^{CTCF-AID}. At day 3 of auxin treatment, 0.75 μM Aphidicolin (left panel) or 2mM H₂O₂ (right panel) was added with or without auxin to the cells for additional 16 hours. At day 4, ESC were fixed and analyzed. Center lines indicate mean values. Percentages of RFP⁺ cells above the threshold are indicated.

2. It would be good to confirm the results of the END-seq with an alternative approach, for instance γH2AX ChIP-seq.

We understand the concern from the reviewer regarding this point. However, END-seq is a more sensitive and specific technique to identify DNA breaks when compared to γH2AX ChIPseq. Moreover, our experimental conditions are not ideal to perform ChIPseq analyses to detect γH2AX as we use asynchronously growing ESC and we believe that the DNA damage might be mostly random in our DUX-induced ESC. Still, we tried and performed γH2AX ChIPseq in untreated and DOX-treated cells for 16 hours (as we did for our END-seq experiment) in the same two clones. However, we didn't feel confident with the results and we could not generate a reliable peak calling as the ratio signal/noise was too high in our samples. However, along with the γH2AX ChIPseq, we also performed a ChIPseq to detect chromatin-bound RPA, also indicative of DNA damage. In this case, we feel more confident about our results and we have included additional data in the manuscript showing increased levels of chromatin-bound RPA at genes and repeats transcriptionally induced by DUX (Fig.1e, f and Supplementary Fig. 3b, c). We did not find significant RPA enrichment at the END-seq sites although this is not surprising if we consider the higher sensitivity of END-seq (estimated in one double strand break per 10,000 cells). This also suggest that DNA breaks at END-seq sites occur in a subset of DUX-induced ESCs. With these new results, we strengthened the idea of DUX-induced transcription as a source for replication stress during 2C-like conversion.

3. The AID-CTCF protein is expressed at significantly lower level than its wild type counterpart. Are ESC homozygous for this allele fully functional, whether for differentiation or for absence of spontaneous 2C-like reprogramming, in spite of reduced CTCF levels? On Fig 3a blot, ZSCAN4 levels seem slightly higher in these than in parental cells (moreover, it is a detail, but there is also a background band that differs between the two lines).

The reviewer is completely right about his/her observations. The CTCF-AID ESC line (52.9.1 as described in the original publication, Nora et al, 2017) shows 2-3-fold-time less level of CTCF compared to the untagged parental cell line. Similarly, our newly generated CTCF-AID ESC lines (Supplementary Fig. 7f) also showed lower levels of CTCF compared to their parental ESC line. This is due to the fact that the AID tag leads to a slight constitutive destabilization of the tagged protein (Morawska and Ulrich, 2013). Despite the lower levels of CTCF, only 72 genes showed differential expression by RNAseq analysis between CTCF-AID and parental ESC lines (Nora et al, 2017). Furthermore, CTCF binding profile in untreated CTCF-AID ESC line were highly similar to wild-type untagged cells (measured by ChIP-exo) highlighting that CTCF tagging does not affect binding (Nora et al, 2017). In addition, CTCF-AID ESC lines differentiate to Neural Stem Cells (NSC) or astrocytes as efficiently as the parental ESC lines (Nora et al, 2017; Nora et al, 2020). We have also confirmed this with our own data (Fig. 4 and Supplementary Fig. 9). Combined, we can conclude that AID-tagging does not interfere with CTCF binding or function. Finally, although we cannot exclude that a very slight increase in the number of 2C-like cells could be detected in untreated CTCF-AID ESC lines compared to their parental cells (Fig. 3b), we did not observe significant increased expression of 2C-genes (Fig. 3g). These minor differences could be also attributable to ESC clone-specific effects (see also new Supplementary Fig. 9a). We also agree with the reviewer that there is a very slight increase in the levels of ZSCAN4 in untreated CTCF-AID ESC lines compared to their parental cells (Fig. 3a), but this increase is negligible compared to the increase observed in CTCF-depleted cells. Moreover, the percentage of ZSCAN4-positive cells is similar in untreated CTCF-AID ESC lines compared to their parental cells (Supplementary Fig. 7b). We are not sure about where is the background band mentioned on the ZSCAN4 blot. The ZSCAN4 antibody we used in our manuscript has been widely used in the literature, we only detect one main protein band in our western blot analyses, and its intensity decreases in shZSCAN4-expressing cells. Thus, we are pretty confident on the specificity of the antibody.

4. Is DNA damage induced upon AID-CTCF degradation?

This is a very important point raised by the reviewer as we would have predicted to observe DNA damage upon CTCF depletion. As previously reported in the original publication (Nora et al, (2017), Fig. S1E), depletion of CTCF does not induce DNA damage evaluated by detecting the levels of γ H2AX.

5. Does CTCF bind close to and directly regulates DUX4?

This an excellent point. The mouse gene DUX or its human version DUX4 are not found as unique copies in the genomes but their ORFs are included within a repeat organized as an array of multiple copies (in humans between 10 and more than a hundred). The human D4Z4 array

(containing DUX4 copies) has been recently sequenced and annotated by Nanopore-based single molecule sequencing (Mitsuhashi et al, 2017). However, the mouse DUX array has not been mapped and the precise sequence of the region containing the DUX genes is unclear. Therefore, it is difficult to assess whether CTCF binds within the DUX array. We observed multiple CTCF binding sites surrounding the DUX array, and these are lost upon CTCF degradation (see below in Figure b). Unfortunately, the region where the DUX array is located is unmapped and noise in the region can be detected even in the input). Nevertheless, our data points to the idea that DUX is not directly regulated by CTCF based on the following: It has been shown that genes directly upregulated upon CTCF depletion tend to be closer to active enhancers than down- or non-regulated genes. In addition, a high fraction of the genes upregulated normally have a TAD boundary separating both, the gene and the enhancer (Nora et al, 2017). This suggests that CTCF repress gene expression by insulating promoter-enhancer interaction through the specification of a TAD boundary. Importantly, we did not detect changes in DUX expression early after CTCF depletion. Indeed, DUX activation is a late event in the 2C-like reprogramming process (Fig. 5a) and thus, is likely to be induced by secondary events and not through direct interaction with a nearby insulated enhancer by CTCF. We mentioned now this in our manuscript (lines 403-405).

Figure b: Genome browser tracks showing ChIPseq RPKM read count at the genomic region (chr10:57,775,049-58,532,580) in the indicated samples. A single-track highlighting identified CTCF peaks in ESC is also shown.

6. Can the authors propose an explanation for the systematically low fraction of cells converting to a 2C-like stage following their various manipulations (e.g. less than 30% upon CTCF degradation, which must presumably occur in all cells)? What is intrinsically different between ZSCAN4- or MERVL-RFP-positive and -negative cells in this experiment?

We understand the reviewer's point. By using CTCF-AID ESC lines we are able to efficiently deplete CTCF to undetectable levels from most ESC in the culture. However, approximately 15-30% of them can be reprogrammed to a 2C-like state four days after depletion. Although we observed a 15-20X increase in the number of LTR-RFP+ ESC, the 2C-like reprogramming phenotype is not fully penetrant. Over the past decade, multiple studies have extensively demonstrated that lineage commitment and cell identity are actively reinforced to resist cell fate changes. The best example of these studies is the somatic cell reprogramming into induced pluripotent stem cells (iPSC). Reprogramming somatic cells to iPSC is a highly inefficient process where only around 1% of fibroblasts successfully acquire pluripotency. The low reprogramming efficiency is due to multiple roadblocks that need to be overcome such as the stoichiometry and

level of the reprogramming factors, specific signaling pathways, microRNA expression, pluripotency-inhibiting transcription factors, DNA methylation and, more importantly, epigenetic modifications that need to be reset to a pluripotent state (reviewed in Brumbaugh et al, 2019). We believe that 2C-like reprogramming mediated by CTCF-depletion has also to overcome certain roadblocks explaining the incomplete reprogramming in all CTCF-depleted cells and the long time necessary for the conversion (3-4 days). To support this idea, we first made the observation that HDAC inhibitors further promote 2C-conversion in CTCF-depleted cells suggesting the existence of epigenetic roadblocks (Supplementary Fig. 7e). In this revised version we also provide the following experiments:

- 1) We first explored whether intrinsic heterogeneity in ESC cultures influences the efficiency of 2C-reprogramming mediated by CTCF-depletion. For this, we first established a total of 23 single-cell derived clonal ESC lines from the parental CTCF-AID ESC. We observed that the endogenous percentage of 2C-like cells within the cultures of these clonal ESC lines varies from 0.17% to 2.87%. Interestingly, the percentage of 2C-like cells observed four days after CTCF depletion in the same clonal ESC lines correlated with the starting percentage. These results suggest that transcriptional and/or epigenetic variation within ESC cultures influences the efficiency of the 2C-like reprogramming and reinforces the idea of existing roadblocks that need to be overcome. This is now in Supplementary Fig. 9a.*
- 2) We also explored whether lineage committed cells derived from ESC showed the same conversion phenotype. Interestingly, Neural Stem Cells (NSC) differentiated from CTCF-AID ESC lines do not undergo 2C-like reprogramming upon CTCF depletion. Major epigenetic and transcriptional changes take place during lineage commitment and differentiation toward NSC. These results suggest that additional roadblocks are established upon differentiation that efficiently impair 2C-like reprogramming in lineage-committed cells. This new piece of data is now in Fig. 4 and Supplementary Fig. 9b-d.*

Finally, as reported by Rodriguez-Terrones et al, 2018, ZSCAN4+MERVL- ESC represent a transcriptionally intermediate state between pluripotent and 2C-like (ZSCAN4+MERVL+) ESC (see also Supplementary Fig. S10).

6. Statistics should be added to plots lacking them (a majority).

In this revised version we have added statistics to all plots.

7. In their model DUX-> de novo transcription/replication conflict -> replicative stress / DNA damage -> 2C conversion, the authors do not explain why it would happen only with DUX and not with other transcriptional activators. As well, they find ZSCAN4 repression to prevent 2C conversion of CTCF-depleted ESC to 2C-like cells, and argue in their discussion that ZSCAN4 could participate in limiting the DNA damage associated with the conversion. What is the link between their observation and this hypothesis?

We understand the reviewer's concern regarding why we did not elaborate on the specific implication of DUX in this phenotype. Compared to most transcription factors, DUX is a

pioneering transcription factor able to bind to nucleosome-occupied DNA. DUX induces major chromatin changes including extensive chromatin opening, de-repression of repetitive sequences and loss of heterochromatin with the final outcome of the transcriptional activation of the 2C-program. Importantly, the 2C transcriptional program is not activated at the same level by any other known transcription factor. Furthermore, in this revised version we now show a direct link between DUX-induced transcriptional activation of the 2C-program and the generation of replication stress (Fig. 1e, f and Supplementary Fig. 3b, c). All these combined might explain why the replication stress phenotype in this experimental setting is specific to DUX and cannot be mimicked by other transcriptional activator.

Regarding ZSCAN4, we and others have demonstrated the relevant role of the ZSCAN4 cluster in the 2C phenotype (Falco et al, 2007; our manuscript). Nevertheless, the role of ZSCAN4 in the 2C-like reprogramming is yet to be precisely determined. We speculated in the discussion that ZSCAN4 could participate in limiting the DNA damage associated with the conversion based on the current literature:

- 1) Transient activation of ZSCAN4 is required for the maintenance of telomeres and genome stability of ESCs (Zalzman et al, 2010).*
- 2) ZSCAN4 binds microsatellite DNA protecting these fragile genomic regions from DNA damage. In fact, ZSCAN4 depletion leads to DNA damage in 2C mouse embryos (Srinivasan et al, 2020).*
- 3) Expression of ZSCAN4 in cultures of ESC becoming aneuploid or polyploid, dramatically increase the number of euploid cells within few days of expression. Surprisingly, it shows the same effect in human primary fibroblasts with Down syndrome (Amano et al, 2015).*
- 4) ZSCAN4 contributes to telomere protection in TRF2-deficient cells (Markiewicz-Potoczny et al, 2021).*

*Interestingly, recent studies in *D. Melanogaster* and *C. elegans* have revealed that transcriptional activation during ZGA generated intrinsic DNA damage that needed to be corrected for proper development (Blythe et al, 2015; Butuci et al, 2015). Thus, due to the reported role of ZSCAN4 in maintaining genome stability and its transient expression at the 2C stage we speculated that ZSCAN4 could limit DNA damage in 2C-like cells. As mentioned above, ZSCAN4 depletion leads to DNA damage in 2C mouse embryos (Srinivasan et al, 2020). Nevertheless, the contribution of ZSCAN4 in limiting DNA damage in 2C-like cells is speculative, only discussed in the Discussion and will need to be explored in further studies. In this revised version, we expanded our reasoning and included additional references (lines 397-403).*

Minor comments:

1. ZSCAN4 is described as one gene all along the paper whereas it is a cluster, with some isoforms DUX4-regulated. This should be clarified.

To avoid any misunderstanding, we have now clarified that ZSCAN4 is considered as a cluster unless otherwise noted.

2. Line 130: ED should be defined once (we suppose the authors mean Embryonic Development).

We have eliminated ED and now reads as “Embryonic development”.

3. ENDseq or END-seq? Either one should be chosen, and then stuck to.

We have now homogenized the naming, and all read as END-seq.

REFERENCES:

Amano, T., Jeffries, E., Amano, M., Ko, A.C., Yu, H. and Ko, M.S. Correction of Down syndrome and Edwards syndrome aneuploidies in human cell cultures. *DNA Research*, 22: 331-342. (2015)

Atashpaz, S., Samadi Shams, S., Gonzalez, J.M., Sebestyén, E., Arghavanifard, N., Gnocchi, A., Albers, E., Minardi, S., Faga, G., Soffientini, P., Allievi, E., Cancila, V., Bachi, A., Fernández-Capetillo, O., Tripodo, C., Ferrari, F., López-Contreras, A.J. and Costanzo, V. ATR expands embryonic stem cell fate potential in response to replication stress. *Elife*. 9, e54756 (2020).

Brumbaugh J, Di Stefano B, Hochedlinger K. Reprogramming: identifying the mechanisms that safeguard cell identity. *Development*. 146: dev182170 (2019).

Butuci, M., Williams, A.B., Wong, M.M., Kramer, B, and Michael, W.M. Zygotic genome activation triggers chromosome damage and checkpoint signaling in *C. elegans* primordial germ cells. *Developmental Cell*, 34: 85-95. (2015)

Blythe, S.A. and Wieschaus E.F. Zygotic genome activation triggers the DNA replication checkpoint at the midblastula transition. *Cell*, 160: 1169-1181. (2015)

Falco, G., Lee, S.L., Stanghellini, I., Bassey, U.C., Hamatani, T. and Ko, M.S. Zscan4: A novel gene expressed exclusively in late 2-cell embryos and embryonic stem cells. *Developmental Biology*, 307: 539-550 (2007).

Markiewicz-Potoczny, M., Lobanova, A., Loeb, A.M., Kirak, O., Olbrich, T., Ruiz, S. and Lazzerini Denchi, E. TRF2-mediated telomere protection is dispensable in pluripotent stem cells. *Nature*, 589: 110-115 (2021).

Mitsuhashi, S., Nakagawa, S., Takahashi Ueda, M., Imanishi, T., Frith, M.C. and Mitsuhashi, H. Nanopore-based single molecule sequencing of the D4Z4 array responsible for facioscapulohumeral muscular dystrophy. *Sci Rep*. 7: 14789. doi: 10.1038/s41598-017-13712-6 (2017).

Morawska, M. and Ulrich, H.D. An expanded tool kit for the auxin-inducible degron system in budding yeast. *Yeast*, 30: 341-51 (2013)

Nora, E.P., Goloborodko, A., Valton, A.L., Gibcus, J.H., Uebersohn, A., Abdennur, N., Dekker, J., Mirny, L.A. and Bruneau, B.G. Targeted Degradation of CTCF Decouples Local Insulation of Chromosome Domains from Genomic Compartmentalization. *Cell* 169, 930-944 (2017).

Nora, E.P., Caccianini, L., Fudenberg, G., So, K., Kameswaran, V., Nagle, A., Uebersohn, A., Hajj, B., Saux, A.L., Coulon, A., Mirny, L.A., Pollard, K.S., Dahan, M. and Bruneau, B.G. Molecular basis of CTCF binding polarity in genome folding. *Nat Commun*, 11: 5612. doi: 10.1038/s41467-020-19283-x (2021).

Srinivasan, R., Nady, N., Arora, N., Hsieh, L.J., Swigut, T., Narlikar, G.J., Wossidlo, M. and Wysocka, J. Zscan4 binds nucleosomal microsatellite DNA and protects mouse two-cell embryos from DNA damage. *Sci Adv*, 6: eaaz9115 (2020).

Storm, M.P., Kumpfmüller, B., Bone, H.K., Buchholz, M., Sanchez Ripoll, Y., Chaudhuri, J.B., Niwa, H., Tosh, D. and Welham, M.J. Zscan4 is regulated by PI3-kinase and DNA-damaging agents and directly interacts with the transcriptional repressors LSD1 and CtBP2 in mouse embryonic stem cells. PLoS One 9, e89821 (2014).

Zalzman, M., Falco, G., Sharova, L.V., Nishiyama, A., Thomas, M., Lee, S.L., Stagg, C.A., Hoang, H.G., Yang, H., Indig, F.E., Wersto, R.P. and Ko, M.S. Zscan4 regulates telomere elongation and genomic stability in ES cells. Nature, 464: 858-863 (2010).

Reviewer #2 (Remarks to the Author):

Olbrich et al examine events following induction of a 2C-like fate from ESCs, which eventually leads to identification of a role for CTCF as a barrier to induction of the 2C state. The authors find 2C induction increases DNA damage, and the sites of damage frequently overlap with binding sites of CTCF and cohesin. CTCF binding is reduced in 2C cells relative to ESCs. Interestingly, depletion of CTCF from ESCs leads to an increase in spontaneous conversion to the 2C state. Finally, the authors show that CTCF depletion leads to upregulation of ZSCAN4 at early time points, which was previously shown to precede emergence of the full 2C phenotype (i.e., MERVL and DUX expression). Overexpression of ZSCAN4 in CTCF depleted cells increased 2C conversion and knock-down of ZSCAN4 blocked reprogramming, suggesting ZSCAN4 is critical for reprogramming to 2C observed in CTCF depleted cells.

The finding that CTCF is a barrier to reprogramming to the 2C state is of considerable interest. By and large, the experiments were well designed and the conclusions well supported. I have a few concerns and several minor points that require attention.

We appreciate the reviewer's enthusiasm about our observations and his/her statement of considerable interest.

Major comments:

1. In Fig. 2a, enrichment is shown at the peak locations specifically from Dox-induced cells. Aggregation specifically at peaks from the Dox+ cells will bias the result. If, for example, the no Dox cells also have numerous END-seq peaks, but these peaks are in different locations than the peaks from Dox+ cells, aggregation over the no Dox peaks would yield the opposite result – that damage is higher without Dox. Peaks from all conditions (Dox+ and Dox-) should be included in any aggregate plot of this type, particularly when the plot is used as an argument that damage is higher in Dox+ cells.

We understand the reviewer's concern. However, there are two important things to consider: 1) DNA damage is greatly induced upon DUX expression in ESC compared to untreated control cells (Figure 1). However, we understand how our sentence could have been misunderstood. Thus, we modified the sentence and now reads as: "DUX-expressing ESC showed de novo accumulation of END-seq signal at specific genomic locations compared to untreated ESC^{Dux}". We only used END-

seq to identify specifically the location of DNA damage occurring recurrently in DOX-treated ESC. Our claim of higher DNA damage in DUX-expressing ESC is strongly supported by other results (γ H2AX Western blots, RPA ChIPseq...). 2) We acknowledge that END-seq peaks can also be detected in untreated cells, likely resulting from the high level of endogenous replication stress found in ESC. Indeed, after performing a peak calling in untreated ESCs we identified 854 overlapping peaks in the two ESC clones used in this study (Supplementary Table 7). Importantly, these peaks do not show overlap with known CTCF binding sites (35 out of 854 or 4%) demonstrating that the newly generated END-seq peaks found in DOX-treated cells are specific to 2C-like conversion and mostly in CTCF-binding sites (718 out of 1539 or 47%).

It is important to consider that END-seq is a very sensitive technique that allowed us to map precisely DNA lesions recurrently occurring in the same genomic location. However, the DNA damage has to occur, and the DNA end generated, recurrently in the same precise genomic location in a high number of cells. In other words, non-recurrent random breaks will be indistinguishable from the background and will generate noise. Thus, it is likely that CTCF-associated DNA damage represent only a fraction of the total DNA damage generated in DUX-expressing ESC.

In this revised version we have now included the peak calling in untreated ESC and expanded this point in the text.

2. Given the somewhat moderate changes in aggregate CTCF enrichment (Fig. 1e), it would be worth normalizing the CUT&RUN data to the contaminating *E. coli* reads (see Meers et al, eLife 2019), which will serve the same purpose as spike-in normalization. Such normalization can flatten differences due to technical variation, such as library quality or read depth.

*We appreciate the reviewer's suggestion as this is something we did not consider initially. We examined the presence of *E. coli* reads in all our Cut&Run experiments and although we did find few of those reads, we did not feel confident enough to use them as a way to normalize our data. Thus, to avoid any misinterpretation, we decided to steer away of claims suggesting global decrease in CTCF binding. Instead, we downsized our datasets to avoid differences due to read depth and performed a very strict peak calling analysis in all our samples. This allowed us to determine that 2C-like cells are characterized by a decrease in the total number of CTCF peaks identified. Indeed, a total of 2662 and 657 overlapping CTCF peaks between two independent ESC^{Dux} clones were lost and gained respectively in 2C-like cells (Fig. 2e-g). Similar trend was observed in endogenous 2C-like cells (Supplementary Fig. 6b, c). These results suggested that 2C-like cells undergo certain reorganization in the CTCF binding landscape mainly characterized by a decrease in the number of CTCF peaks identified.*

3. The gene expression data in Fig. S6c suggest that CTCF loss results in partial (but incomplete) induction of the 2C transcription pattern. (Comparing DUX-expressing LTR+ cells to auxin-treated cells shows weaker induction of 2C genes in CTCF depleted cells than in DUX expressing cells.) However, a more direct comparison would be LTR+ auxin-treated cells to LTR+ DUX-expressing cells, which was not made. In addition, the other figures describing gene expression changes upon CTCF depletion, Figures 3c, 3d, and 3g, show differences in expression of 30, 6 (in browser tracks), and 10 genes, respectively. Why are variable subsets of 2C genes quantified from experiment to experiment rather than showing the same 40+ 2C-specific genes (from S6c) in 3c

and 3g? It is particularly strange to select 30 genes for one plot and 10 for another plot in the same figure.

There are several points in this comment, and we will try to explain our reasoning:

- 1) *As shown in Fig. 3b, we observed between 15-30% of $MERVL^+$ cells (depending on the experiment) 4 days after CTCF depletion. Importantly, the RNAseq datasets used in Supplementary Fig. 7c correspond to data obtained from total RNA isolated from unsorted auxin-treated $ESC^{CTCF-AID}$ (Nora et al, 2017). Thus, at day 4 there is 70-85% of cells that are not efficiently converted to a 2C-like state contributing to the perception that 2C-reprogramming is partial or incomplete (we now mention this in the figure legend). To demonstrate that this is not the case we included below (Figure c) a real-time PCR analysis for the detection of several 2C-markers in total auxin-treated $ESC^{CTCF-AID}$ or $MERVL^+$ cells sorted from auxin-treated $ESC^{CTCF-AID}$. This analysis clearly show that levels of these 2C-associated genes is greater in $MERVL^+$ sorted cells compared to total auxin-treated $ESC^{CTCF-AID}$. Moreover, we do not think is completely fair to compare DUX-expressing cells to endogenous $MERVL^+$ cells sorted from auxin-treated $ESC^{CTCF-AID}$. Based on our own results, DUX and thus, other 2C-genes induced by DUX, in doxycycline-inducible systems like ours, could reach expression levels above physiological compared to spontaneously induced 2C-converted ESC.*

Figure c: Graphs showing the relative fold change (log2) expression of five 2C-associated genes in untreated, auxin-treated for four days, untreated but sorted and auxin-treated for four days but sorted for RFP (LTR-RFP reporter) $ESC^{CTCF-AID}$. Real-time PCR reactions were performed by triplicate. GAPDH expression was used to normalize gene expression.

- 2) *Supplementary Fig. 6c shows the expression level of the 50 genes with the highest differential expression change within the samples and includes many 2C genes. However, it also includes genes that are downregulated upon DUX expression and not considered canonical 2C genes. We believed that in this case, building the heatmap with differentially expressed genes instead of only canonical 2C genes would show better the clustering of the different samples.*
- 3) *Fig. 3c shows the expression level of 30 2C-genes extracted from RNAseq datasets. It is an arbitrary number determined by us to provide higher significance to the dataset as this is*

the first plot showing 2C-like conversion from a transcriptional point of view from CTCF-depleted ESC. It is simple to extract a wish list of 30 genes from RNAseq datasets. Importantly, a similar plot to the one shown in Fig. 3c can also be obtained by only using five 2C genes (Figure d below). These genes, included in all plots, (Fig. 5b or Fig. 3g) are a representative subset of 2C-genes and accurately correlate with the 2C-like state. With the addition of few more genes in these plots, all individually confirmed by real-time PCR, we just wanted to add more significance to the dataset. We could validate additional genes in Fig. 5b or add more examples (tracks) to Fig 3d, but the plots won't show significant differences in the trend compared to the ones included.

Figure d: Graph showing the relative fold change (\log_2) expression of a subset of five representative 2C-associated genes (DUX, ZSCAN4, ZFP352, SP110, TDPOZ1 and PRAMEL7) in LTR-RFP reporter ESC^{CTCF-AID} untreated or treated with auxin for four days or washed off for additional two days. Data obtained from Nora et al, 2017.

Minor points:

1. Fig. 1b: In these images, the RFP+ cells are substantially reduced at the longest time points, which is inconsistent with the quantification in 1a. If the images are not representative of the population quantified in 1a, it is not clear why they are included.

We completely agree with the reviewer in this point. We selected one field with one colony to pursue a better detail, but it was not completely representative. In this revised version, we changed this series of images by one showing a different field area where it is now possible to see 2C-like converted cells 48 hours after the addition of doxycycline (Supplementary Fig. 2a). In the Supplementary Video 1 included it is possible to observe in a wider field the 2C-like conversion dynamics.

2. A more thorough discussion of the END-seq data would be helpful, including a brief mention of how single and double stranded lesions are identified, and the significance of each class of lesion to the interpretation of what type of DNA damage may be inducing these breaks.

We agree with the reviewer's point as more information about the END-seq data and our interpretation will help to understand better our observations. In this revised version, we have expanded the discussion of the END-seq data including how single and double stranded lesions are identified (see Methods), and the significance of each class of lesion to understand the

phenotype (see Discussion). We don't feel we can include more information regarding the type of damage inducing these lesions as this is currently unknown and one of the goals of our collaborators using ENDseq. Stalled replication forks, reverse forks or certain type of repeats can create single ends but unclear yet in our case.

3. The authors discuss the reduction in CTCF binding upon induction of 2C, but there did not seem to be any mention of whether novel CTCF binding sites emerge in 2C cells. It is worth reporting the number (if any) of new CTCF binding sites found in 2C cells and discussing these in the text.

We thank the reviewer for this important point, which we did not initially consider. As mentioned above, we performed a very strict peak calling analysis in all our samples. This allowed us to determine that 2C-like cells are characterized by a decrease in the total number of CTCF peaks identified. Indeed, a total of 2662 and 657 overlapping CTCF peaks between two independent ESC^{Dux} clones were lost and gained respectively in 2C-like cells (Fig. 2e-g).

4. "...2C-like reprogramming was further boosted cooperatively by expressing low levels of DUX or by incubating ESCCTCF-AID with HDAC inhibitors, known to promote 2C-like conversion..." A more thorough description (beyond this one sentence) of the alternative 2C reprogramming methods (low levels of DUX or HDACi) would be helpful.

We appreciate the reviewer's comment and following his/her advice we have extended and included a more thorough description of the two methods promoting 2C-like conversion by which we examined their cooperation with the loss of CTCF in ESC^{CTCF-AID}.

5. "Collectively, these results demonstrated that chromatin bound CTCF prevents 2C-like conversion." It is not clear why "chromatin bound" is included in this sentence, as it is not strictly true that depletion of the "chromatin bound" fraction was shown to cause the phenotype.

We agree with the reviewer and removed "chromatin-bound" from the sentence.

6. "Furthermore, over-expression of ZSCAN4C boosted 2C-like conversion as early as 24 hours specifically in CTCF-depleted ESC while cells with normal levels of CTCF did not show major changes in the number of 2C-like cells (Fig. 4d and Extended data Fig. 8e)." This conclusion appears to be inaccurate, as overexpression of ZSCAN4C also increases 2C-like cells in the parental cells (Fig. 4d). I do not see this as a concern about the data, but the figure should be described accurately.

We thank the reviewer for this observation as our sentence was truly inaccurate and we did not properly convey to the reader our results. In this revised version, we have corrected this sentence.

7. The "b" and "c" labels for S8b and S8c seem to be swapped (or alternatively their descriptions in the legend are swapped). In addition, the immunofluorescence data in S8c would be stronger if the fraction of cells expressing ZSCAN4, LTR-RFP, or both were quantified.

We thank the reviewer for this observation, and we have corrected it. In addition, we have now

quantified the immunofluorescence from Supplementary Fig. 8c (now 10c) and included a new plot showing this quantification in Supplementary Fig. 10d.

8. Several details are missing from the methods and/or figure legends. For example, the amount of Dox used in each experiment and the amount of CTCF antibody used for CUT&RUN need to be reported.

We thank the reviewer for this observation and included the aforementioned missing details in the corresponding Figure Legends or Methods.

REFERENCES:

Nora, E.P., Caccianini, L., Fudenberg, G., So, K., Kameswaran, V., Nagle, A., Uebersohn, A., Hajj, B., Saux, A.L., Coulon, A., Mirny, L.A., Pollard, K.S., Dahan, M. and Bruneau, B.G. Molecular basis of CTCF binding polarity in genome folding. Nat Commun, 11: 5612. doi: 10.1038/s41467-020-19283-x (2021).

Reviewer #3 (Remarks to the Author):

Olbrich et al. show a surprising role for CTCF in preventing the switch from pluripotency to totipotency. Totipotency or 2C-like state is characterized by the upregulation of endogenous retroviruses. The authors show that inducible expression of the transcription factor DUX activate genes that are characteristic of the 2C-like state. Associated with this state are DNA breaks which overlap with the position of CTCF sites. In the 2C-like cells CTCF and CTCF binding is decreased, which is consistent with CTCF levels in 2C-stage embryos. When CTCF is acutely depleted, cells also upregulate genes associated with the 2C-like state. The exact mechanism behind this remains unclear, however, the authors show that knock-down of Zscan4, which is a target of DUX mitigates the induction of the 2C-like state as a consequence of CTCF depletion.

The manuscript presents an exciting and unexpected role for CTCF in preventing reprogramming to the totipotent state. I believe the manuscript is well presented and balanced. I recommend this paper for publication. I have a few suggestions to improve the manuscript.

We appreciate the reviewer's general view about our manuscript and his/her positive recommendation for publication.

* The authors discuss the decrease they observe in CTCF binding in terms of a "more relaxed chromatin architecture" (p.7 line 144). However, no evidence is shown for this and this pure conjecture. I believe it is better to steer away from these types of conclusion or show actual data. This comment is further complicated by the fact that severe decrease in CTCF can have rather

mild effect on chromatin looping. How CTCF would lead to more relaxed chromatin is unknown and not discussed, as such it remains an interesting correlation between relaxed chromatin and the absence of CTCF.

We appreciate the reviewer's comment and agree with his/her point of view. In this revised version we steered away from those subjective terms and focus on more tangible observations. We performed a very stringent peak calling analysis on our Cut&Run data and observed consistent loss of and gain of subsets of CTCF peaks in 2C-like cells suggesting certain reorganization in the CTCF binding landscape upon 2C-like reprogramming. Whether this leads to a more relaxed chromatin architecture in ESC needs to be further demonstrated and it is not precisely the goal of our manuscript.

* Nora et al have shown that the direct effects of CTCF depletion on gene expression are likely to be genes that have CTCF bound near their promoters. Have the authors considered that CTCF loss may lead to a decrease in the expression of repressors of the Dux gene or other regulator of the 2C-like state.

We appreciate the reviewer's comment as this is something we should have considered. We have examined the expression level of different reported regulators of the 2C transition (including Chaf1a, Cha1b, Rif1, Kdm1a, Trim28, Ehmt2, Setdb1, Pogf6, Ythdc1 or Smchd1 (Macfarlan et al, 2012; Rodriguez-Terrones et al, 2018; Wu et al, 2020; Liu et al, 2021; Huang et al, 2021) 24 and 48 hours after CTCF depletion (Supplementary Fig. 11a). A clear change in the expression of these genes should have been observed at these timepoints if CTCF had a direct effect in their regulation (CTCF bound at their promoters or blocking enhancer/promoter interactions). However, we did not detect significant expression changes in these genes at those timepoints (Supplementary Fig. 11a). This is in contrast to the expression of ZSCAN4a which changes quickly upon CTCF depletion.

* Is there anything particular to the CTCF sites that overlap with the peaks in the END-seq data? Are they loop anchors for instance?

We appreciate the reviewer's comment as this is something we considered but we did not include in the final version of the manuscript. We thoroughly analyzed whether there was something particular about the CTCF sites overlapping END-seq signal. In particular, this is the analysis we performed to detect enrichment for loop anchors and domain boundaries:

Among 764 END-Seq peaks overlapped with +/- 5kb of CTCF peaks:

*122 are only in loop anchors (15.96%)
59 are both in loop anchors and domain boundaries
108 are in domain boundaries
475 are no located neither of them*

From a random selection of 764 CTCF peaks:

*126 are only in loop anchors (16.49%)
92 are both in loop anchors and domain boundaries
133 are in domain boundaries*

413 are not located neither of them

From this analysis we concluded there is no enrichment in loop anchors or domain boundaries in the CTCF/END-seq sites. The most significant enrichment we found is what we finally included in the manuscript: “More than 25% of the END-seq peaks localized within a 10kb distance from a DUX binding site. Furthermore, 16% of the 1220 genes associated by proximity to END-seq peaks, including well-known 2C genes, were strongly upregulated by DUX (Supplementary Fig. 4d and Supplementary Tables 4, 5)”. Our new RPA ChIPseq data strongly support our hypothesis regarding the implication of DUX-induced transcription on generating replication stress and DNA damage (Fig. 1e, f and Supplementary Fig. S3b, c).

*** Why is it surprising that knock-down of Zscan4 abrogates reprogramming to the 2C-like state? Is this not exactly why the experiment was performed?**

We completely agree with the comment. As the reviewer mentioned, we performed the experiment to confirm whether ZSCAN4 downregulation abrogated 2C-like conversion. We have now changed the sentence as we indeed validated our hypothesis.

*** The effects on mRNA levels following reconstitution of CTCF (wash-off) are relatively mild. The authors could consider to do a nascent RNA sequencing experiment, which may yield a stronger regression to the pluripotent transcription state.**

We agree with the reviewer’s suggestion to consider a nascent RNA sequencing experiment to determine whether a stronger regression could be detected. However, there are several points to be aware of for this experiment which makes it difficult to pursue: 1) For a nascent RNaseq experiment it is necessary a high number of starting cells (probably several millions of ESCs). Because we have to sort live LTR-RFP⁺ ESC (between 30-50% of total positive cells) for all the conditions of the experiment, it would be technically challenging. 2) More importantly, we also performed live cell imaging on a similar experimental setting as the one shown in Figure 3g and noticed that even though CTCF expression was restored, the fate for most of the RFP⁺ ESC was eventually cell death as shown in Supplementary Fig. 2e. Therefore, we face a very small temporal window to examine changes in gene expression upon restoration of CTCF expression before the cells die. Of note, in the experiment shown in Fig. 3g, we only sorted live LTR-RFP⁺ ESC. Still, considering these limitations, we believe there is a significant change in the gene expression of a representative subset of 2C genes due to CTCF re-expression suggesting a direct role for CTCF in regulating this transcriptionally program. Importantly, in this experimental setting, CTCF is re-expressed only for few hours.

*** It was unclear whether the Zscan4 shRNAs target only Zscan4c or other copies as well. Are the other copies not upregulated as well? Please discuss this in more detail in the main text.**

We did not make this clear in the manuscript. The shRNA has a perfect sequence match with the isoforms ZSCAN4C, D and F, one mismatch with ZSCAN4A and two mismatches with ZSCAN4B. We included this information in the Methods. As also mentioned in the text, although all ZSCAN4 transcripts are expressed in ESC, ZSCAN4C and ZSCAN4F are the most abundant (Storm et al, 2014) and with a perfect match for the shRNA.

* The authors mention “stabilize the 2C-like state” (p.11 line 224). This seem like a counter-intuitive statement. The 2C stage is transient, therefore there may be no regulatory loop to stabilize the 2C stage and by extension the 2C-like state.

We agree with the reviewer. We have changed the sentence by “promote the 2C-like state”.

REFERENCES:

Macfarlan, T.S., Gifford WD, Driscoll S, Lettieri K, Rowe HM, Bonanomi D, Firth A, Singer O, Trono D, Pfaff SL. Embryonic stem cell potency fluctuates with endogenous retrovirus activity. *Nature* **487**, 57-63 (2012).

Rodriguez-Terrones, D., Gaume, X., Ishiuchi, T., Weiss, A., Kopp, A., Kruse, K., Penning, A., Vaquerizas, J.M., Brino, L. and Torres-Padilla, M.E. A molecular roadmap for the emergence of early-embryonic-like cells in culture. *Nat. Genet.* **50**, 106-119 (2018).

Wu, K., Liu, H., Wang, Y., He, Y., Xu, S., Chen, Y., Kuang, J., Liu, J., Guo, L., Li, D., Shi, R., Shen, L., Wang, Y., Zhang, X., Wang, J., Pei, D. and Chen, J. SETDB1-Mediated Cell Fate Transition between 2C-Like and Pluripotent States. *Cell Rep.* **30**, 25-36 (2020).

Liu, J., Gao, M., He, J., Wu, K., Lin, S., Jin, L., Chen, Y., Liu, H., Shi, J., Wang, X., Chang, L., Lin, Y., Zhao, Y., Zhang, X., Zhang, M., Luo, G., Wu, G., Pei, D., Wang, J., Bao, X. and Chen, J. The RNA m⁶A reader YTHDC1 silences retrotransposons and guards ES cell identity. *Nature* **591**: 322-326 (2021).

Huang, Z., Yu, J., Cui, W., Johnson, B.K., Kim, K. and Pfeifer, G.P. The chromosomal protein SMCHD1 regulates DNA methylation and the 2c-like state of embryonic stem cells by antagonizing TET proteins. *Sci Adv*, doi: 10.1126/sciadv.abb9149 (2021).

Storm, M.P., Kumpfmüller, B., Bone, H.K., Buchholz, M., Sanchez Ripoll, Y., Chaudhuri, J.B., Niwa, H., Tosh, D. and Welham, M.J. Zscan4 is regulated by PI3-kinase and DNA-damaging agents and directly interacts with the transcriptional repressors LSD1 and CtBP2 in mouse embryonic stem cells. *PLoS One* **9**, e89821 (2014).

REVIEWERS' COMMENTS

Reviewer #1 (Remarks to the Author):

Congratulation for going the distance in addressing thoroughly my comments. An excellent paper, a very interesting observation.

Didier Trono

Reviewer #2 (Remarks to the Author):

Although most of my concerns were satisfied, two major concerns were not.

1. (Point #1 in initial review) The concern raised was that quantification of read enrichment of Dox+ and Dox- cells over peaks called from Dox+ cells is biased. Hypothetically, if END-seq was performed on two identical sets of cells, peaks were called from set #1, and read density of sets #1 and set #2 were plotted over set #1 peaks, set #1 will likely show higher read enrichment due simply to sampling error, which is often unavoidable. If two sets of random data were generated and peaks were called from one, the same thing would be observed (high enrichment of random data set A over its own peaks, low enrichment of random data set B over peaks from A). Therefore, the plot in Figure 2A is not biologically meaningful.

In response, the authors raised a number of points indicating DNA damage is higher +Dox and that the peaks in -Dox cells do not significantly overlap CTCF binding sites. These points are fine but they do not address the fundamental concern raised with regards to Figure 2A. I suggest the authors plot these data in an unbiased manner (as suggested in the previous review) or remove this plot. Furthermore, inclusion of the peaks from -Dox cells in the Supplementary Data is probably not needed, since these peaks are not analyzed in any meaningful way in the manuscript and are not important for any conclusions.

2. (Point #3 in initial review) The major concern was with the series of comparisons of 2C gene expression levels upon CTCF depletion or depletion followed by add-back. Here, the authors used different subsets of 2C genes for each comparison, without any clear justification (Figs. 3c, 3g, S7c). In the rebuttal, the authors state "Fig. 3c shows the expression level of 30 2C-genes extracted from RNAseq datasets. It is an arbitrary number determined by us to provide higher significance to the dataset as this is the first plot showing 2C-like conversion from a transcriptional point of view from CTCFdepleted ESC. It is simple to extract a wish list of 30 genes from RNAseq datasets. [...] With the addition of few more genes in these plots, all individually confirmed by real-time PCR, we just wanted to add more significance to the dataset."

Although I appreciate the authors' veracity in their explanation, it is bad practice (and non-scientific) to cherry-pick a different set of genes for each figure on the basis of "add[ing] more significance" for each comparison. Considering that there appears to be several dozen genes expressed in 2C cells but not ES cells (S7C), it seems a simple matter to generate an extensive, unbiased, and well-justified list of 2C marker genes for the comparisons in Figures 3c and 3g. This would add credibility to the conclusions made about the effect of CTCF loss on the 2C state.

Reviewer #3 (Remarks to the Author):

The author have addressed all my comments.

“More than 25% of the END-seq peaks localized within a 10kb distance from a DUX binding site. Furthermore, 16% of the 1220 genes associated by proximity to END-seq peaks, including well-known 2C genes, were strongly upregulated by DUX (Supplementary Fig. 4d and Supplementary Tables 4, 5)”

I recommend the authors place these enrichments in the context of the expected overlap from a random selection.

POINT-BY-POINT RESPONSE TO THE REVIEWERS' COMMENTS

We first would like to thank the reviewers for the last round of comments. I truly believe that our manuscript has greatly improved from our first version. Please find below a point-by-point response to all the concerns and suggestions made by the reviewers.

REVIEWERS' COMMENTS

Reviewer #1 (Remarks to the Author):

Congratulation for going the distance in addressing thoroughly my comments. An excellent paper, a very interesting observation.

Didier Trono

We sincerely thank the reviewer for this comment. It is really gratifying to read this from you.

Reviewer #2 (Remarks to the Author):

Although most of my concerns were satisfied, two major concerns were not.

1. (Point #1 in initial review) The concern raised was that quantification of read enrichment of Dox+ and Dox- cells over peaks called from Dox+ cells is biased. Hypothetically, if END-seq was performed on two identical sets of cells, peaks were called from set #1, and read density of sets #1 and set #2 were plotted over set #1 peaks, set #1 will likely show higher read enrichment due simply to sampling error, which is often unavoidable. If two sets of random data were generated and peaks were called from one, the same thing would be observed (high enrichment of random data set A over its own peaks, low enrichment of random data set B over peaks from A). Therefore, the plot in Figure 2A is not biologically meaningful.

In response, the authors raised a number of points indicating DNA damage is higher +Dox and that the peaks in -Dox cells do not significantly overlap CTCF binding sites. These points are fine but they do not address the fundamental concern raised with regards to Figure 2A. I suggest the authors plot these data in an unbiased manner (as suggested in the previous review) or remove this plot. Furthermore, inclusion of the peaks from -Dox cells in the Supplementary Data is probably not needed, since these peaks are not analyzed in any meaningful way in the manuscript and are not important for any conclusions.

The reviewer struggles with the Figure 2A and suggests removing it. Although the reviewer argues that this observation could be random, we analyzed two independent clones with a significant overlap in the peaks identified in Dox-treated samples over Dox-untreated samples which questions the randomness of our observations. Still, I understand how this plot could be misinterpreted and thus, in this revised version we have eliminated the plot from the figure and the peaks from Dox-untreated cells as suggested by the reviewer.

2. (Point #3 in initial review) The major concern was with the series of comparisons of 2C gene expression

levels upon CTCF depletion or depletion followed by add-back. Here, the authors used different subsets of 2C genes for each comparison, without any clear justification (Figs. 3c, 3g, S7c). In the rebuttal, the authors state “Fig. 3c shows the expression level of 30 2C-genes extracted from RNAseq datasets. It is an arbitrary number determined by us to provide higher significance to the dataset as this is the first plot showing 2C-like conversion from a transcriptional point of view from CTCFdepleted ESC. It is simple to extract a wish list of 30 genes from RNAseq datasets. [...] With the addition of few more genes in these plots, all individually confirmed by real-time PCR, we just wanted to add more significance to the dataset.”

Although I appreciate the authors’ veracity in their explanation, it is bad practice (and non-scientific) to cherry-pick a different set of genes for each figure on the basis of “add[ing] more significance” for each comparison. Considering that there appears to be several dozen genes expressed in 2C cells but not ES cells (S7C), it seems a simple matter to generate an extensive, unbiased, and well-justified list of 2C marker genes for the comparisons in Figures 3c and 3g. This would add credibility to the conclusions made about the effect of CTCF loss on the 2C state.

We reason the selection of the genes included in the figures 3c, 3g and S7c) as follows:

-We wrote that “Figure S7C shows the expression level of the 50 genes with the highest differential expression change within the samples and includes many 2C genes. However, it also includes genes that are downregulated upon DUX expression and not considered canonical 2C genes. We believed that in this case, building the heatmap with differentially expressed genes instead of only canonical 2C genes would show better the clustering of the different samples.” In this plot we just wanted to show that CTCF-depleted cells clustered with DUX-induced ESC and for this, we included the top 50 most differentially expressed genes upon Dux expression. Therefore, this plot is completely unbiased. As it can also be observed in FigS7C most of these genes are Gm(x), genes largely uncharacterized, and these genes are not commonly used in the field to establish the identity of the 2C-like cells.

*-Instead, for Fig 3c and Fig 3g as well as for all other graphs from the manuscript showing the expression of 2C-associated genes upon CTCF-depletion, we displayed the genes and repeats that are commonly used as markers associated to a 2C-like state in similar publications from the field (Hendrikson et al, Nature Genetics **49**, 925-934; Rodriguez-Terrones et al, Nature Genetics **50**, 106-119; Eckersley-Maslin et al, Genes and Dev **33**, 194-208, just to name a few). What is important is that all these genes are differentially expressed in the 2C embryo compared to ICM/ESC. Furthermore, just as an example, in a very recent paper from two weeks ago (Iturbide et al, Nature Molecular Structural Biology, doi: 10.1038/s41594-021-00590-w) the authors used only the LTR-based reporter and Zscan4 as readout to identify 2C-like cells. As it seems that the homogenization of these two plots (Fig 3c and Fig 3g) is the real concern of the reviewer, we have re-analyzed the data from Nora et al, Cell **169**, 930-944, to include the MERVL repeats and downsized the graph in 3C to include only the 10 genes/repeats used in the other graphs (see new Fig3c). I consider that these genes/repeats (DUX, ZSCAN4, ZFP352, TCSTV3, SP110, TDPOZ1, DUB1 (USP17IA), EIF1ad8 (GM8300), PRAMEL7 and MERVL repeats) represent a list of extensive and consolidated 2C-associated markers in the field (see the Figure below to confirm the validity of these markers and the publications mentioned above). In summary, I find unnecessary to analyze the expression of dozens of additional genes or create a new list of 2C-like genes as these markers are completely valid and commonly used in the field. A new analysis with different genes will not add more credibility to our conclusions as we clearly demonstrated that CTCF loss resulted in 2C-like conversion with the same markers used in similar publications to ours.*

(A) Averaged relative fold change (\log_2) expression of the genes/repeats (DUX, ZSCAN4, ZFP352, TCSTV3, SP110, TDPOZ1, DUB1, EIF1a, PRAMEL7 and MERVL repeats) in Zygotes, Early 2C, Late 2C, Inner Cell Mass (ICM) of blastocysts and ESC. Data obtained from Wu et al, Nature **534**, 652-657. Dux is transiently expressed only in the early 2C embryo. (B) Averaged relative fold change (\log_2) expression of the genes (DUX, ZSCAN4, ZFP352, TCSTV3, SP110, TDPOZ1, DUB1, EIF1a and PRAMEL7) in the same samples from Fig S7C. MERVL repeats are not included as ESC^{DUX} cells are sorted based on the LTR-GFP reporter.

Reviewer #3 (Remarks to the Author):

The author have addressed all my comments.

“More than 25% of the END-seq peaks localized within a 10kb distance from a DUX binding site. Furthermore, 16% of the 1220 genes associated by proximity to END-seq peaks, including well-known 2C genes, were strongly upregulated by DUX (Supplementary Fig. 4d and Supplementary Tables 4, 5)”

I recommend the authors place these enrichments in the context of the expected overlap from a random selection.

We sincerely thank the reviewer for this comment. In the final revised version, we have included the reviewer’s recommendations.